# FEDERATED FLOW MATCHING

## ABSTRACT

Data today is decentralized, generated and stored across devices and institutions where privacy, ownership, and regulation prevent centralization. This motivates the need to train generative models directly from distributed data locally without central aggregation. In this paper, we introduce Federated Flow Matching (FFM), a framework for training flow matching models under privacy constraints. Specifically, we first examine FFM-vanilla, where each client trains locally with independent source and target couplings, preserving privacy but yielding curved flows that slow inference. We then develop FFM-LOT, which employs local optimal transport couplings to improve straightness within each client but lacks global consistency under heterogeneous data. Finally, we propose FFM-GOT, a federated strategy based on the semi-dual formulation of optimal transport, where a shared global potential function coordinates couplings across clients. Experiments on synthetic and image datasets show that FFM enables privacy-preserving training while enhancing both the flow straightness and sample quality in federated settings, with performance comparable to the centralized baseline.

## 1 INTRODUCTION

Generative models (Sohl-Dickstein et al., 2015; Ho et al., 2020; Song et al., 2020) aim to capture the probability distribution of complex data such as images, audio, or text, enabling the synthesis of realistic new samples. Flow matching (FM) provides a powerful framework for this task (Lipman et al., 2022; Albergo & Vanden-Eijnden, 2022; Liu et al., 2022). It learns a deterministic vector field that continuously transports a simple source distribution $q_0$ (e.g., Gaussian noise) to a complex target distribution $q_1$ (e.g., natural images). Training proceeds by regressing the model velocity to a prescribed target velocity defined along paths between paired samples $(x_0, x_1)$ with $x_0 \sim q_0$ and $x_1 \sim q_1$. The choice of coupling between $q_0$ and $q_1$ is fundamental, as it dictates the geometry of probability flows. Independent couplings, formed by pairing $x_0$ and $x_1$ at random, are straightforward but induce curved probability paths. These curved paths require many integration steps during sampling and thus slow inference. Flow matching based on optimal transport (OT) (Tong et al., 2020; Onken et al., 2021; Liu, 2022; Tong et al., 2023) selects couplings that minimize transport cost, often by solving mini-batch OT problems in training. The resulting flows are straighter and allow fewer steps at inference, significantly accelerating generation.

Most existing generative methods assume that all data is centralized. In practice, this is often not the case. Data are created and stored on personal devices (Yang et al., 2019), and across national boundaries where privacy, ownership, and regulation preclude direct sharing. The question, then, is how to train a single generative model across these dispersed sources while keeping raw data local. Federated learning (FL) (Konečný et al., 2016b; McMahan et al., 2017) provides a paradigm in which each client performs training on its own data and communicates only model updates or gradients to a coordinating server. The server aggregates these updates into a global model and redistributes them back to the clients. In this way, raw data never leaves local storage, while knowledge is shared through the iterative exchange of model parameters. FL is well-established for machine learning problems, but its extension to flow matching has, to our knowledge, not been explored.

To this end, we introduce Federated Flow Matching (FFM) for training flow matching models based on decentralized data. FFM follows the standard FL paradigm (see Fig. 1): clients keep their data on-device and compute local updates to a shared flow model, while a central server aggregates these updates via federated averaging without ever accessing raw training samples. The unique challenge in FFM, compared to standard FL, lies in constructing effective

couplings between the source distribution ($q_0$) and the aggregated target distribution ($q_{1,\lambda} = \sum_i \lambda_i q_1^i$). Here, $q_1^i$ is client $i$'s data distribution and $\lambda_i$ is its aggregation weight. Unlike the centralized setting, which allows for the direct computation of such couplings, the federated setting prohibits combining isolated client data. The key difficulty is, therefore, how to construct effective couplings from decentralized data and learn a global velocity field that not only preserves privacy but also yields straight flows necessary for fast inference.

We start with the baseline method FFM-vanilla, in which each client treats source and target distributions independently, yielding the coupling $\sum_i \lambda_i (q_0 \otimes q_1^i)$ [1], where $q_0$ is the common source distribution. This baseline guarantees privacy and allows collaborative training through federated averaging. While the resulting flows may be curved, FFM-vanilla establishes a simple and effective foundation for decentralized generative modeling.

We then introduce FFM-LOT, wherein each client $i$ computes the local OT plan $\pi_i^*$ between $q_0$ and $q_1^i$. The coupling is constructed as an aggregation of local OT plans defined as $\sum_i \lambda_i \pi_i^*$. This approach improves flow straightness within each client's data region (Fig. 2 (c)), thereby enhancing inference efficiency. However, under heterogeneous data distributions, the aggregated local OT plans may yield curved trajectories and fail to achieve global straightness.

Next, we propose FFM-GOT that directly approximates the global OT plan across all clients to strengthen the flow straightness. By leveraging the semi-dual formulation of OT, FFM-GOT learns a shared dual potential function that implicitly coordinates couplings across clients without data sharing. Both the dual potential and the flow model are updated during training via federated averaging. This approach enables the learning of globally straight probability flows (Fig. 2 (c)), significantly improving inference efficiency while maintaining strict privacy constraints.

Lastly, experiments on synthetic and image benchmarks show that all these methods enable effective training under privacy constraints. Specifically, FFM-vanilla provides a simple and stable baseline. FFM-LOT improves inference efficiency by leveraging local OT plans to straighten probability paths, but is sensitive to non-Independent and Identically Distributed (non-IID) client distributions. FFM-GOT achieves the highest inference efficiency by learning globally straight paths, albeit at the cost of more computation. Remarkably, FFM-GOT outperforms the centralized method OT-CFM (Tong et al., 2023) at a low number of function evaluations (NFE). This is because FFM-GOT directly approximates the global OT plan, whereas OT-CFM relies on mini-batch approximations.

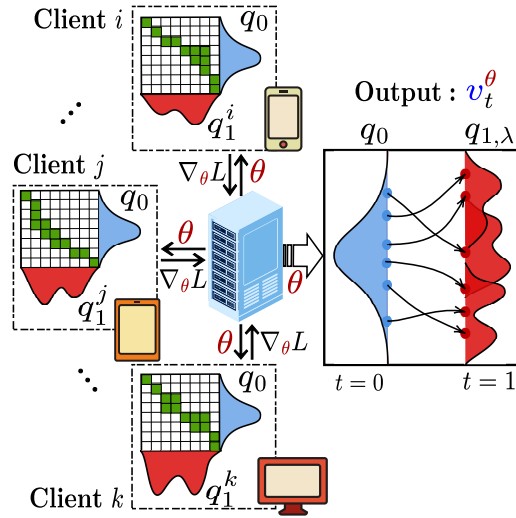

Figure 1: Federated Flow Matching (FFM). Each client holds its own data distribution and shares a common base distribution. The server aggregates client updates to learn a shared vector field that transports the base distribution toward the federated mixture, while raw data remain local.

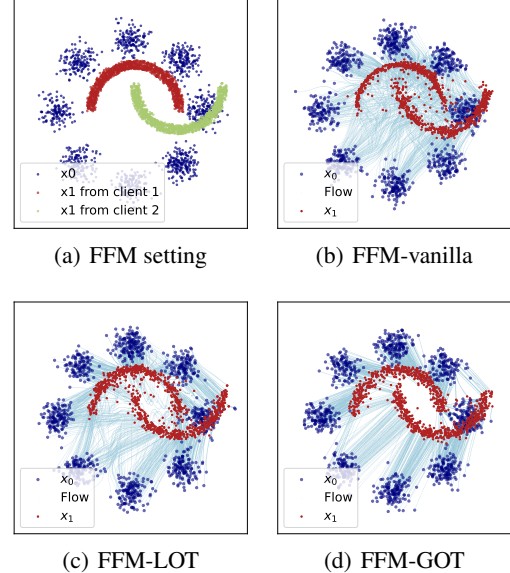

Figure 2: Visualization on 2D benchmark.

---

[1] Here $q_0 \otimes q_1^i$ denotes the product measure

## 2 PRELIMINARIES AND PROBLEM FORMULATION

### 2.1 OPTIMAL TRANSPORT PROBLEM: KANTOROVICH AND DYNAMIC FORMULATION

OT provides a principled way to measure the cost of transforming one probability distribution into another. Given two probability measures $q_0$ and $q_1$ on $\mathbb{R}^d$ and a measurable cost function $c : \mathbb{R}^d \times \mathbb{R}^d \to \mathbb{R}$, the Kantorovich formulation of OT is

$$\text{OT}_c(q_0, q_1) = \min_{\pi \in \Pi(q_0, q_1)} \int_{\mathbb{R}^n \times \mathbb{R}^n} c(x_0, x_1) \, d\pi(x_0, x_1), \tag{1}$$

where $\Pi(q_0, q_1)$ is the set of couplings with marginals $q_0$ and $q_1$. Intuitively, $\pi$ specifies how mass from $q_0$ is transported to $q_1$, and the objective seeks the plan of minimal cost. For the quadratic cost $c(x_0, x_1) = \frac{1}{2}\|x_0 - x_1\|^2$, the OT cost equals the squared 2-Wasserstein distance, $\mathcal{W}_2^2(q_0, q_1)$. Benamou and Brenier (Benamou & Brenier, 2000) showed that this admits a dynamic reformulation

$$\mathcal{W}_2^2(q_0, q_1) = \min_{(q_t, v_t)} \left\{ \int_0^1 \int_{\mathbb{R}^d} \|v_t(x)\|^2 \, q_t(x) \, dx \, dt \ \middle| \ \begin{matrix} \partial_t q_t + \nabla \cdot (q_t v_t) = 0, \\ q_{t=0} = q_0, \ q_{t=1} = q_1. \end{matrix} \right\}, \tag{2}$$

which seeks a time-dependent vector field $v_t$ that generates a probability flow $q_t$ transporting $q_0$ to $q_1$ with minimal kinetic energy. At optimality, $p_t$ traces the displacement interpolation between $q_0$ and $q_1$, meaning that for $(x_0, x_1) \sim \pi^*$, the OT plan, the trajectory is given by $x_t = (1-t)x_0 + tx_1$ with constant velocity $v_t(x_t) = x_1 - x_0$. Thus, the solution corresponds to straight-line paths between optimally coupled points (Villani, 2003).

### 2.2 FLOW MATCHING

Flow matching aims to sample from the target distribution $q_1$ by transforming samples from the source distribution $q_0$. The particle dynamics follow the ordinary differential equation (ODE) $dx_t = u_t(x_t) \, dt$, which induces a probability flow $q_t$ with densities evolving according to the continuity equation $\partial_t p_t + \nabla \cdot (p_t u_t) = 0$. Since the exact vector field $u_t$ is intractable, it is approximated by a neural network $v_t^\theta$, which can be trained via the conditional flow matching objective

$$\mathcal{L}_{\text{CFM}}(q_0, q_1; \theta) = \mathbb{E}_{t \in \mathcal{U}[0,1], (x_0, x_1) \sim \pi} \left\| v_t^\theta((1-t)x_0 + tx_1) - (x_1 - x_0) \right\|^2, \quad \pi \in \Pi(q_0, q_1) \tag{3}$$

where $\Pi(q_0, q_1)$ is the set of all joint distributions having marginal distributions $q_0$ and $q_1$. As shown by Pooladian et al. (2023), for any admissible coupling $\pi \in \Pi(q_0, q_1)$, perfect training of minimizing (3) yields a vector field that generates a valid flow between $q_0$ and $q_1$. Different choices of the coupling distribution $\pi$ lead to different flow matching methods. For instance, independent coupling flow matching (I-CFM) sets $\pi = q_0 \otimes q_1$, where $x_0$ and $x_1$ are sampled independently. This simple construction is effective but typically produces curved trajectories, requiring many integration steps. Optimal Transport-based flow matching (OT-CFM) refines this approach by setting $\pi = \pi^\star$, the optimal transport plan from (1). The resulting training problem becomes

$$\mathcal{L}_{\text{OT-CFM}}(\theta) = \mathbb{E}_{t \in \mathcal{U}[0,1], (x_0, x_1) \sim \pi^*} \left\| v_t^\theta((1-t)x_0 + tx_1) - (x_1 - x_0) \right\|^2.$$

In this case, training aligns $v_t^\theta$ with the displacement interpolations of the Benamou–Brenier problem (2), producing straight trajectories that reflect the geodesic structure of Wasserstein space and enabling faster, more stable inference. However, computing $\pi^*$ has cubic computational complexity in the number of samples, which is challenging for large datasets. Alternatively, one can approximate $\pi^*$ using mini-batch data (Tong et al., 2023) or use entropic OT solvers (Pooladian et al., 2023; Klein et al., 2025).

### 2.3 PROBLEM FORMULATION

We consider the problem of training a single flow matching model in a federated learning setting. Suppose that there are $n$ clients and each client $i$ possesses a local data distribution $q_1^i$, which cannot be shared with other clients due to the privacy constraints. The global target distribution is defined as $q_{1,\lambda} = \sum_{i=1}^n \lambda_i q_1^i$, where $\lambda_i$ is the weight for each client $i$ and satisfies $\sum_{i=1}^n \lambda_i = 1$. We assume that all clients share a common source distribution $q_0$.

The objective of federated flow matching is to learn a global vector field $v_t^\theta$ that transports $q_0$ to the aggregated target $q_{1,\lambda}$ by minimizing

$$\mathcal{L}_{\text{FFM}}(\theta) = \mathbb{E}_{t \in \mathcal{U}[0,1], (x_0, x_1) \sim \pi} \left\| v_t^\theta((1-t)x_0 + tx_1) - (x_1 - x_0) \right\|^2, \quad \pi \in \Pi(q_0, q_{1,\lambda}). \quad (4)$$

The difficulty is that the coupling $\pi$ is defined with respect to the global target distribution $q_{1,\lambda}$, which cannot be formed without centralizing client data. The goal of this work is to overcome this challenge by developing federated learning algorithms that learn from decentralized data and yield efficient, straight flows.

# 3 FEDERATED FLOW MATCHING

## 3.1 VANILLA FEDERATED FLOW MATCHING (FFM-VANILLA)

The performance of conditional flow matching depends critically on the choice of coupling $\pi$ in (4). A natural baseline is to take $\pi$ as the independent product measure $\pi = q_0 \otimes q_{1,\lambda}$, which corresponds to sampling $x_0 \sim q_0$ and $x_1 \sim q_{1,\lambda}$ independently. This leads to the federated flow matching objective in its vanilla form, i.e.,

$$\mathcal{L}_{\text{FFM-vanilla}}(\theta) = \mathbb{E}_{t, x_0 \sim q_0, x_1 \sim q_{1,\lambda}} \left\| v_t^\theta((1-t)x_0 + tx_1) - (x_1 - x_0) \right\|^2$$

$$= \sum_{i=1}^n \lambda_i \, \mathbb{E}_{t, x_0 \sim q_0, x_1 \sim q_1^i} \left\| v_t^\theta((1-t)x_0 + tx_1) - (x_1 - x_0) \right\|^2, \quad (5)$$

whereby the global objective in (5) is decomposed into a weighted sum of client-specific expectations. Each term in the sum depends exclusively on the local data distribution, making it directly implementable with standard FL. This formulation allows each client to compute gradients using only its local data, which are then aggregated on a central server through weighted averaging.

The federated optimization procedure is outlined in Algorithm 1. In each communication round, client i samples a mini-batch of data pairs $(x_0, x_1)$ with $x_0 \sim q_0$ and $x_1 \sim q_1^i$. For each pair, it draws $t \sim \mathcal{U}[0, 1]$, forms the interpolation $x_t = (1-t)x_0 + tx_1$, evaluates the local loss $L^i$. Then, it computes the stochastic gradient $\nabla_\theta L^i$ and sends it to the server. The server aggregates client updates by a weighted average $\sum_{i=1}^n \lambda_i \nabla_\theta L^i$ and applies the global update $\theta \leftarrow \theta - \eta_\theta \sum_{i=1}^n \lambda_i \nabla_\theta L^i$ with learning rate $\eta_\theta$.

FFM-vanilla provides privacy guarantees, as clients communicate only model gradients rather than raw samples $x_1 \sim q_1^i$. It is simple to implement and provides a natural baseline

---

**Algorithm 1** FFM-vanilla

1: **Input**: Source distribution $q_0$, weight vector $\lambda$, data distribution $q_1^i$, $i = 1, \ldots, n$,
2: **for** $k = 0, \ldots, K$ **do**
3:   **for** each client $i = 1, \ldots, n$ **do**
4:     Sample $x_0 \sim q_0$, $x_1 \sim q_1^i$, and $t \sim \mathcal{U}[0, 1]$
5:     Update $x_t = (1-t)x_0 + tx_1$ and compute loss $L^i \leftarrow \left\| v_t^\theta(x_t) - (x_1 - x_0) \right\|^2$
6:     Send gradient $\nabla_\theta L^i$ to server
7:   **end for**
8:   Server update $\theta \leftarrow \theta - \eta_\theta \sum_{i=1}^n \lambda_i \nabla_\theta L^i$
9: **end for**
10: **Output**: Global velocity field $v_t^\theta$.

---

for federated flow matching. However, its reliance on independent couplings might lead to highly curved probability paths. During inference, integrating the ODE defined by the learned vector field $v_t^\theta$ requires numerous evaluation steps to maintain accuracy, resulting in computationally expensive and slow sampling. Thus, although FFM-vanilla succeeds in learning a generative model for the aggregated distribution $q_{1,\lambda}$, it does not achieve efficient inference, motivating the more advanced methods introduced in the rest of this section.

## 3.2 FEDERATED FLOW MATCHING VIA LOCAL OT (FFM-LOT)

It is shown by Tong et al. (2023) that one would greatly benefit from training flow models by using the optimal coupling (joint distribution) in the Kantorovich optimal transport rationale. Training with the OT plan yields a vector field that solves the Benamou-Brenier dynamic OT formulation, producing straight probability paths and enabling fast inference. However, directly computing the global OT plan between $q_0$ and $q_{1,\lambda}$ is impossible and incompatible with federated learning, as it

requires access to the entire decentralized dataset. Herein, we explore methods to approximate the benefits of OT within the federated learning constraints.

A natural initial strategy is to compute an OT plan locally on each client between the shared source $q_0$ and its local target $q_1^i$. The global coupling is then constructed as the mixture of these local OT plans, $\pi_{\text{local}}^* = \sum_{i=1}^n \lambda_i \pi_i^*$, where $\pi_i^* = \arg\min_{\pi_i \in \Pi(q_0, q_1^i)} \int_{\mathbb{R}^n \times \mathbb{R}^n} c(x_0, x_1) \, d\pi_i(x_0, x_1)$. This leads to the following federated learning objective

$$\mathcal{L}_{\text{FFM-LOT}}(\theta) = \mathbb{E}_{t \sim \mathcal{U}[0,1], (x_0, x_1) \sim \pi_{\text{local}}^*} \left\| v_t^\theta((1-t)x_0 + tx_1) - (x_1 - x_0) \right\|^2$$

$$= \sum_{i=1}^n \lambda_i \, \mathbb{E}_{t \sim \mathcal{U}[0,1], (x_0, x_1) \sim \pi_i^*} \left\| v_t^\theta((1-t)x_0 + tx_1) - (x_1 - x_0) \right\|^2. \quad (6)$$

The federated algorithm for this approach, detailed in Algorithm 2, proceeds as follows. In each communication round, each client $i$ samples a mini-batch of $x_0 \sim q_0$ and $x_1 \sim q_1^i$. Each client $i$ computes the OT plan $\hat\pi_i$ from this pair of mini-batch data, which can be achieved by exact or approximate (entropic regularized via Sinkhorn) OT solvers. It then samples pairs $(x_0, x_1)$ from this plan, computes the flow matching loss, and sends the gradient to the server for aggregation. The server then aggregates client gradients and performs a global update.

Using locally optimal couplings, FFM-LOT encourages straighter paths within each client's data distribution, which can lead to faster inference times compared to the vanilla independent coupling. However, the aggregate of local OT

---

**Algorithm 2** FFM-LOT

1: **Input**: Source distribution $q_0$, weight vector $\lambda$, data distribution $q_1^{i=1,\ldots,n}$
2: **for** $k = 0, \ldots, K$ **do**
3:     **for** client $i = 1, \ldots, n$ **do**
4:         Sample $x_0 \sim q_0$, $x_1 \sim q_1^i$, and $t \in \mathcal{U}[0,1]$,
5:         Resample $(x_0, x_1) \sim \pi_i^* = \text{OT}(x_0, x_1)$
6:         Updated $x_t = (1-t)x_0 + tx_1$ and compute loss $L^i \leftarrow \left\| v_t^\theta(x_t) - (x_1 - x_0) \right\|^2$
7:         Send gradient $\nabla_\theta L^i$ to server
8:     **end for**
9:     Server updates $\theta \leftarrow \theta - \eta_\theta \sum_{i=1}^n \lambda_i \nabla_\theta L^i$ and broadcast to clients
10: **end for**
11: **Output**: Global velocity field $v_t^\theta$.

---

plans $\pi_{\text{local}}^*$ is generally not equivalent to the global OT plan $\pi^*$. The local approach fails to account for the geometric relationships between data points across different clients. Consequently, the resulting vector field is a compromise that averages these local, potentially conflicting, optimal trajectories rather than finding a truly globally optimal flow. The following theorem quantifies the sub-optimality gap between the mixture of local plans and the true global plan, highlighting its dependence on the statistical heterogeneity between the client distributions. The proof of Theorem 1 is given in Appendix A.4.

**Theorem 1 (Sub-optimality of mixed local OT plans).** *Let* $\pi_{\text{local}}^* = \sum_{i=1}^n \lambda_i \, \pi_i^*$*, and*

$$\pi_i^* = \underset{\pi_i \in \Pi(q_0, q_1^i)}{\operatorname{argmin}} \int c(x_0, x_1) \, d\pi_i(x_0, x_1), \quad \pi^* = \underset{\pi \in \Pi(q_0, q_{1,\lambda})}{\operatorname{argmin}} \int c(x_0, x_1) \, d\pi(x_0, x_1).$$

*Suppose that the Monge setting* $\pi^* = (\text{Id}, T^*)_\# q_0$ *and* $\pi_i^* = (\text{Id}, T_i^*)_\# q_0$ *for* $i = 1, \ldots, n$*. Assume that* $\|T^*(x)\|, \|T_i^*(x)\| \le D$ *for* $q_0$*-a.e.* $x$*. Under the standard regularity condition, there exists* $C > 0$ *such that* $\mathcal{W}_2^2(\pi^*, \pi_{\text{local}}^*) \le C D \sum_{i=1}^n \lambda_i \, \mathcal{W}_2(q_1, q_1^i)^{2/15}$*.*

Theorem 1 confirms that the sub-optimality of the local OT approach is directly proportional to the average Wasserstein distance between the global target distribution and each client's local distribution. In highly non-IID settings where clients have disparate data, this error can be significant, limiting the inference efficiency of the learned model. This limitation motivates the need for a method that can more directly approximate the *global* OT plan in a federated manner, which we address in the following section.

## 3.3 FEDERATED FLOW MATCHING VIA GLOBAL OT (FFM-GOT)

To further improve flow straightness, we propose a method that directly approximates the global optimal transport plan in a federated manner. Our innovation is based on the semi-dual formulation of the Kantorovich problem, which is presented in the following lemma.

**Lemma 1** (**Kantorovich duality (Villani, 2003; Peyré & Cuturi, 2019)**). *Let $q_0, q_{1,\lambda}$ be probability measures on $\mathbb{R}^n$ and let $c : \mathbb{R}^n \times \mathbb{R}^n \to \mathbb{R}$ be a cost function. The Kantorovich optimal transport problem admits the dual formulation*

$$\mathrm{OT}_c(q_0, q_{1,\lambda}) = \max_{f,g \in L^1} \left\{ \int_{\mathbb{R}^n} f(x_0) \, dq_0(x_0) + \int_{\mathbb{R}^n} g(x_1) \, dq_{1,\lambda}(x_1) \right\} \tag{7}$$

*subject to $f(x_0) + g(x_1) \leq c(x_0, x_1)$ for all $(x_0, x_1)$. The c-transform of $f$ is defined as $f^c(x_1) = \min_{x_0 \in \mathbb{R}^n} \{ c(x_0, x_1) - f(x_0) \}$, which yields the semi-dual representation (Choi et al., 2023)*

$$\mathrm{OT}_c(q_0, q_{1,\lambda}) = \max_{f \in L^1} \left\{ \int_{\mathbb{R}^n} f(x_0) \, dq_0(x_0) + \int_{\mathbb{R}^n} f^c(x_1) \, dq_{1,\lambda}(x_1) \right\}. \tag{8}$$

Using Lemma 1 and the definition of $q_{1,\lambda}$, we obtain

$$\mathrm{OT}_c(q_0, q_{1,\lambda}) = \max_{f \in L^1(q_0)} \left\{ \int f(x_0) \, dq_0(x_0) + \sum_{i=1}^m \lambda_i \int f^c(x_1) \, dq_1^i(x_1) \right\}$$

$$= \max_{f \in L^1(q_0)} \sum_{i=1}^m \lambda_i \left[ \int f(x_0) \, dq_0(x_0) + \int f^c(x_1) \, dq_1^i(x_1) \right]. \tag{9}$$

This reformulation reveals that the global OT problem can be expressed as a federated optimization over the shared dual potential $f$. Once the optimal potential $f^*$ is obtained, the optimal coupling $\pi^*$ can be recovered: A pair $(x_0, x_1)$ is coupled under $\pi^*$ if and only if it satisfies the condition $f^*(x_0) + f^{*c}(x_1) = c(x_0, x_1)$. In practice, we can approximate a sample from $\pi^*$ by first sampling $x_1 \sim q_1$ and then solving $\bar{x}_0 = \arg\min_{x_0} \{ c(x_0, x_1) - f^*(x_0) \}$, after which we use the pair $(\bar{x}_0, x_1)$.

Based on this insight, we propose to learn the dual potential function in a FL paradigm to coordinate couplings among clients. To this end, we parameterize the dual potential $f$ with a neural network $f_\phi$ and optimize the semi-dual objective collaboratively across clients, yielding the following federated learning objective:

$$\mathcal{L}_{\mathrm{FFM\text{-}GOT}}(\theta; \phi) = \sum_{i=1}^n \lambda_i \, \mathbb{E}_{t \sim \mathcal{U}[0,1], (x_0, x_1) \sim \hat{\pi}_\phi} \left\| v_t^\theta((1-t)x_0 + tx_1) - (x_1 - x_0) \right\|^2, \tag{10}$$

where $\hat{\pi}_\phi$ is an empirical approximation of the global OT plan based on the current dual potential $f_\phi$. As the dual potential $f_\phi$ converges towards the global optimum, the resampling procedure provides increasingly accurate approximations of pairs from the true global OT plan $\pi^*$, which in turn allows the flow matching model $v_t^\theta$ to learn straighter, globally optimal paths.

FFM-GOT employs a two-stage optimization process executed over federated communication rounds, as detailed in Algorithm 3. The procedure iteratively updates the vector field and the dual potential. For the vector field update, each client samples a mini-batch of $x_0 \sim q_0$ and $x_1 \sim q_1^i$. Using the current dual potential $f_\phi$, each client then resamples coupled pairs $(x_0, x_1)$ via Algorithm 5. This resampling step effectively identifies, for each target point $x_1$, a corresponding source point that minimizes $c(x_0, x_1) - f_\phi(x_0)$ among $K$ candidate source samples. Based on these coupled pairs, each client computes the gradient $\nabla_\theta L_i$ for the vector field and transmits it to the server for aggregation. The update of the dual potential, detailed in Algorithm 4, proceeds as follows. Each client $i$ com-

---

**Algorithm 3** FFM-GOT

1: **Input**: Source distribution $q_0$, weight vector $\lambda$, data distribution $q_1^i$, $i = 1, \ldots, n$
2: **for** $k = 0, \ldots, K$ **do**
3:     **for** client $i = 1, \ldots, n$ **do**
4:         Sample $x_0 \sim q_0$, $x_1 \sim q_1^i$, and $t \in \mathcal{U}[0,1]$,
5:         Resample $(x_0, x_1) \sim \hat{\pi}_\phi$ via Alg. 5
6:         Update $x_t = (1-t)x_0 + tx_1$ and compute loss $L_\theta^i \leftarrow \left\| v_t^\theta(x_t) - (x_1 - x_0) \right\|^2$
7:         Send gradient $\nabla_\theta L_\theta^i$ to server
8:     **end for**
9:     Server updates $\theta \leftarrow \theta - \sum_{i=1}^n \lambda_i \nabla_\theta L_\theta^i$ and broadcast it to clients
10:     Call DUALUPDATE$(q_0, \{q_1^i\}, \lambda; \phi)$ via Alg. 4
11: **end for**
12: **Output**: Global velocity field $v_t^\theta$ and dual potential function $f_\phi$.

---

putes the loss $L_\phi^i = f_\phi(x_0) + f_\phi^c(x_1)$ on its local data, evaluates the gradient $\nabla_\phi L_\phi^i$, and sends this gradient to the server. The server then aggregates these client gradients and updates the global dual potential network $f_\phi$ using federated averaging.

Note that two approximations cannot be avoided when training flow models using FFM-GOT. The first arises in Alg. 4 when evaluating the c-transform $f_\phi^c$, which requires solving the optimization problem: $\inf_{x_0} c(x_0, x_1) - f(x_0)$. In practice, this infimum can be approximated using a finite steps of gradient descent or existing solvers. The second approximation occurs in Alg. 5 during the resampling of pairs $(x_0, x_1) \sim \hat{\pi}_\phi$, where $\hat{\pi}_\phi$ is an empirical approximation of the global OT plan based on the current dual potential $f_\phi$. This resampling is typically performed over a finite set of candidate source points, which also introduces discretization errors. In practice, our experiments demonstrate that high-quality approximations can be achieved without significantly increasing the computational overhead, allowing FFM-GOT to remain both practical and efficient. More ablation studies on the approximations can be found in Appendix C.3.

**Remark 1.** *In the dual formulation* (7)*, one could parameterize* $g(x_1)$ *and define its c-transform function* $g^c(x_0)$*. However, this c-transform function requires minimizing over* $x_1$*, which is generally more challenging given the multi-modal nature of typical target distributions. In contrast, our chosen parameterization defines the dual potential on the target space, so its c-transform requires minimization only over* $x_0$*. Since* $x_0$ *is usually from a simple, unimodal reference distribution, this minimization is computationally more tractable.*

**Remark 2** (**Schrödinger bridge flow matching**). *The Schrödinger bridge (Shi et al., 2023; Chen et al., 2021) seeks the most likely coupling relative to a prior* $R_\varepsilon(dx, dy) = q_0(dx)p_\varepsilon(dy \mid x)$ *while enforcing marginals* $q_0$ *and* $q_{1,\lambda}$*. Its dual reduces to a single-potential semi-dual,* $\max_\psi \left\{ \int \psi \, dq_{1,\lambda} - \int \log\left( \int e^{\psi(y)} p_\varepsilon(dy \mid x) \right) q_0(dx) \right\}$*, which differs from the OT semi-dual only by the entropic log-sum-exp. Thus, the Schrödinger version of flow matching can be trained in exactly the same manner, with privacy preserved by the decomposition* $\int \psi \, dq_{1,\lambda} = \sum_{i=1}^n \lambda_i \int \psi \, dq_1^i$*. In the zero-noise limit* $\varepsilon \to 0$*, this formulation converges to the classical Kantorovich semi-dual.*

## 4 RELATED WORKS

**Flow matching:** Our work builds upon the recent advances in flow-based generative models, specifically flow matching (FM) (Lipman et al., 2022), which provide a simple and stable framework for training continuous normalizing flows. A key design choice in FM is the coupling between source and target points. The naive independent coupling leads to curved probability paths and slow inference. Recent breakthroughs have significantly improved this by integrating ideas from optimal transport. Notably, Tong et al. (2023) and Pooladian et al. (2023) demonstrated that using mini-batch optimal transport couplings results in straighter paths and much faster sampling. Moreover, Klein et al. (2025); Calvo-Ordonez et al. (2025) showed that entropic regularization of OT couplings can enhance numerical stability and computational efficiency. These methods, however, are designed for centralized data. We generalize this line of work to the FL setting, where data cannot be pooled for mini-batch or entropic OT computation. To address this constraint, we propose a novel federated algorithm that approximates global OT plans across clients without sharing raw data.

**Federated Learning for Generative Models:** Federated Learning (FL) was originally proposed to enable collaborative learning from data distributed across multiple clients without sharing raw data (Konečný et al., 2016a; McMahan et al., 2017). Its application to generative modeling is more recent and introduces unique challenges because the goal is to learn the full complex data distribution. Previous research has explored adapting various generative frameworks to the federated setting. Initial work focused on Federated Generative Adversarial Networks (GANs) (Augenstein et al., 2020), which employ an adversarial minmax game to learn a direct mapping from noise to data samples. This approach stands in contrast to flow-based methods, as GANs do not involve continuous normalizing flows or ODE integration, thereby avoiding the specific inference efficiency challenges associated with neural ODE solvers. More recent efforts have investigated federated diffusion models (Peng et al., 2025; de Goede et al., 2024; Vora et al., 2024). While both diffusion models and flow matching can be formalized as methods that learn a probability path between noise and data distributions, they differ fundamentally in their flexibility. Diffusion models are constrained to a fixed probability path determined by a predefined forward process, whereas flow

matching methods can have from infinitely many possible paths depending on the coupling between source and target distributions. This flexibility enables the design of the probability path geometry for improved inference efficiency, which is not possible in diffusion models. Our work is the first, to our knowledge, to address federated learning of flow matching models, with a specific focus on leveraging optimal transport to achieve fast sampling.

**Optimal Transport:** Its application in machine learning is vast, including generative modeling (Arjovsky et al., 2017; Salimans et al., 2018), domain adaptation (Courty et al., 2017), distribution comparison (Wang et al., 2023), and more recently, as a core component in training flow matching models (Tong et al., 2023; Kornilov et al., 2024; Rohbeck et al., 2025; Wang et al., 2025; Corso et al., 2025; Klein et al., 2024). The semi-dual formulation of OT has been explored for scalable computation (Genevay et al., 2016) and recently for generative modeling in the centralized setting (Choi et al., 2023). Our federated OT method is inspired by these works but addresses the fundamentally different challenge of optimizing the semi-dual objective without sharing data across clients, using a federated averaging procedure on the parameters of a dual potential network. To the best of our knowledge, this is the first work to propose and analyze a federated algorithm for learning OT maps via the semi-dual formulation.

## 5 EXPERIMENTS

### 5.1 ILLUSTRATIVE 2D EXAMPLE

We begin the evaluation on low-dimensional synthetic datasets. We train the vector fields for two pairs of two-dimensional datasets (source distribution $\rightarrow$ target distribution): 8Gaussian $\rightarrow$ moon, and uniform $\rightarrow$ 8Gaussian. In the federated setting, we simulate $n = 2$ clients. The target distribution is partitioned such that each client holds a disjoint subset of the target samples. The source distribution $q_0$ is shared across both clients. Further details on the data splitting are provided in Appendix C.1. We measure sample quality using the Wasserstein distance between the true target distribution and the generated distribution. We also report inference time as a function of NFE during ODE integration.

As shown in Fig. 3, FFM-GOT consistently outperforms both FFM-vanilla and FFM-LOT across all NFEs, achieving lower Wasserstein distances with fewer integration steps. Visualized results are shown in Figs. 2 and 4. FFM-GOT produces nearly straight trajectories between source and target samples, while FFM-vanilla exhibits curved and inefficient paths. FFM-LOT improves over the vanilla method but still falls short of global optimality due to client heterogeneity. This confirms that FFM-GOT learns straighter probability paths, enabling faster and more accurate sampling.

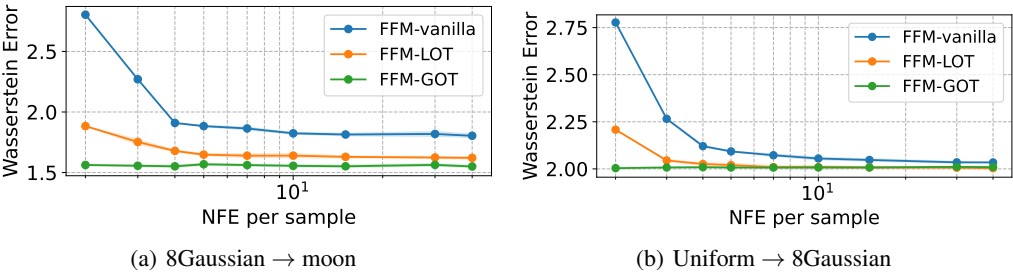

(a) 8Gaussian $\rightarrow$ moon        (b) Uniform $\rightarrow$ 8Gaussian

Figure 3: Generation performance of different federated flow matching models on different NFEs.

### 5.2 IMAGE GENERATION

To evaluate the performance of the proposed methods, we conduct unconditional image generation on CIFAR and Imagenet.

We start with CIFAR and simulate $n = 3$ clients. Tables 1 and 2 report the Fréchet Inception Distance (FID) and Inception Score (IS), respectively, achieved by the three methods under varying numbers of function evaluations (NFEs) and degrees of data heterogeneity. FFM-vanilla provides a simple but effective baseline. However, it performs poorly at small NFEs due to curved probability

paths. FFM-LOT improves straightness of the learned flows, as evidenced by better generation performance at small NFEs. However, it shows sensitivity to data heterogeneity, as its performance advantage over FFM-vanilla diminishes under the more non-IID setting ($\alpha = 0.1$). This aligns with Theorem 1, confirming that the suboptimality of aggregated local OT plans increases with client distribution divergence. FFM-GOT achieves superior performance at low NFEs, demonstrating its ability to learn globally straight path that enables efficient generation. As shown in Table 2, it also attains the best IS across all tested NFEs, reflecting improved diversity in generated images. At high NFEs, the performance gap between these methods narrows since all methods converge to high performance. However, FFM-GOT degrades at high NFEs due to approximation errors from the dual potential function. Specifically, two sources of approximation accumulate during long integration: (1) the c-transform is optimized with only a limited number of gradient steps, and (2) the argmin over $x_0$ is restricted to a finite candidate pool. These biases are small for a small value of NFE, but would accumulate with many discretization steps. These results highlight the context-dependent strengths of each approach: FFM-GOT for few-step and diverse generation, FFM-LOT for high-quality convergence in moderate non-IID settings, and FFM-vanilla for simplicity and stability.

Remarkably, FFM-GOT demonstrates superior performance at low NFEs compared to the centralized baseline OT-CFM (Tong et al., 2023). As shown in Fig. 5, FFM-GOT surpasses this centralized baseline (FID 28.6 at NFE=3) after 160K training steps. We attribute this advantage to our direct approximation of the global OT plan. In contrast, centralized OT-CFM relies on mini-batch OT approximations, which may not fully capture the global data geometry, especially with limited batch sizes. Furthermore, FFM-LOT performs competitively with (even better than) centralized OT-CFM using either Euler (Table 1) or dopri5 (Table 7) integration solvers. This is notable because FFM-LOT computes OT plans locally on each client's data, and the aggregated local plans serve as a proxy for the global coupling. The overall effective sample size used across all clients is larger than the single batch used in centralized OT-CFM, which may contribute to its superior performance. In other words, the performance gain from having a larger sample size outweighs the performance degradation introduced by federation. These performance gains come with increased computational overhead, as detailed in Table 5. More experimental results, including the ablation study on the c-transform approximation and the visualization of generated samples, are provided in the Appendix C.

Table 1: FID across NFE values of CIFAR dataset with 3 clients.

| NFE | Federated: Dirichlet $\alpha = 0.3$ | | | Federated: Dirichlet $\alpha = 0.1$ | | | Centralized |
|-----|-------------|---------|---------|-------------|---------|---------|---------|
|     | FFM-vanilla | FFM-LOT | FFM-GOT | FFM-vanilla | FFM-LOT | FFM-GOT | OT-CFM |
| 3   | 57.46 | 29.58 | **18.15** | 50.57 | 30.16 | **24.97** | 28.60 |
| 10  | 16.23 | 11.84 | **8.37** | 13.28 | 12.14 | **8.75** | 12.07 |
| 20  | 9.37 | 7.76 | **7.41** | 7.82 | 8.05 | **7.52** | 8.09 |
| 50  | 6.36 | **5.38** | 7.65 | **5.26** | 5.53 | 7.68 | 5.57 |
| 100 | 5.26 | **4.42** | 8.04 | **4.34** | 4.50 | 8.07 | 4.55 |

Table 2: Inception score across NFE values of CIFAR dataset with 3 clients.

| NFE | Federated: Dirichlet $\alpha = 0.3$ | | | Federated: Dirichlet $\alpha = 0.1$ | | | Centralized |
|-----|-------------|---------|---------|-------------|---------|---------|---------|
|     | FFM-vanilla | FFM-LOT | FFM-GOT | FFM-vanilla | FFM-LOT | FFM-GOT | OT-CFM |
| 3   | 1.90 | 2.49 | **3.08** | 1.96 | 2.47 | **3.02** | 9.01 |
| 10  | 3.27 | 3.84 | **4.75** | 3.47 | 3.82 | **4.69** | 9.00 |
| 20  | 4.15 | 4.61 | **5.47** | 4.42 | 4.59 | **5.41** | 9.02 |
| 50  | 4.75 | 5.11 | **5.92** | 5.11 | 5.11 | **5.88** | 9.00 |
| 100 | 4.96 | 5.29 | **6.10** | 5.29 | 5.28 | **5.98** | 8.98 |

To illustrate the scalability of our methods in large-scale federated settings, we simulate $n = 32$ clients on CIFAR. In each iteration, 3 clients are randomly selected to compute gradients and communicate with the server. The per-iteration training time remains comparable to the setup with fewer clients, as it is primarily determined by the number of clients participating per round. In Table 3, we report the FID for Dirichlet parameters $\alpha = 0.5$ and $\alpha = 0.1$, and the results align with our

Table 3: FID across NFE values of CIFAR dataset with 32 clients.

| NFE | Federated: Dirichlet $\alpha = 0.5$ | | | Federated: Dirichlet $\alpha = 0.1$ | | | Centralized |
| | FFM-vanilla | FFM-LOT | FFM-GOT | FFM-vanilla | FFM-LOT | FFM-GOT | OT-CFM |
| --- | --- | --- | --- | --- | --- | --- | --- |
| 3 | 53.50 | 31.48 | **23.40** | 54.25 | 30.92 | **25.42** | 28.60 |
| 10 | 14.29 | 12.33 | **8.78** | 14.53 | 12.09 | **9.32** | 12.07 |
| 20 | 8.18 | 8.14 | **7.78** | 8.33 | 8.03 | **7.97** | 8.09 |
| 50 | **5.47** | 5.59 | 7.96 | **5.49** | 5.63 | 8.12 | 5.57 |
| 100 | **4.47** | 4.53 | 8.30 | **4.64** | 4.72 | 8.43 | 4.55 |

previous findings. The strategy of selecting a subset of clients per round accelerates training without degrading performance, confirming that our methods scale effectively with the number of clients.

We further evaluate generative performance on ImageNet64-500, a more challenging benchmark constructed from 500 ImageNet classes, each containing 500 samples, yielding a total of 250,000 images. We simulate $n = 4$ clients. The details of federated data setup is provided in Appendix C.4.1. We train models with three methods for 200K steps. The FID scores are presented in Table 4. The results show that FFM-GOT achieves the best performance at NFE=4 and FFM-LOT performs best across other NFE values. The advantage of FFM-GOT at low NFE is less pronounced here, likely because the high-dimensional data space makes learning the corresponding dual potential function more challenging.

We further evaluate generative performance on ImageNet64-500, a benchmark constructed from 500 ImageNet classes, each containing 500 samples, yielding a total of 250,000 images. We simulate $n = 4$ clients. The details of federated data setup is provided in Appendix C.4.1. All models, including the three FFM variants and the centralized OT-CFM baseline, are trained for 200K steps. The FID scores in Table 4 highlight the difficulty of this task, as even the centralized OT-CFM baseline struggles to achieve strong results. However, our FFM methods perform better possibly due to large effective batch sizes. Among federated methods, FFM-GOT performs best at NFE=4, while FFM-LOT achieves the highest scores at other NFE values. The advantage of FFM-GOT at low NFE is less pronounced here, likely because the high-dimensional data space makes learning the corresponding dual potential function more challenging.

Table 4: FID across NFE values of Imagenet dataset with 4 clients.

| NFE | Federated: Dirichlet $\alpha = 0.3$ | | | Centralized |
| | FFM-vanilla | FFM-LOT | FFM-GOT | OT-CFM |
| --- | --- | --- | --- | --- |
| 4 | 64.3 | 43.7 | **39.4** | 65.2 |
| 10 | 29.2 | **18.6** | 28.4 | 29.1 |
| 20 | 24.8 | **16.3** | 23.2 | 28.7 |
| 50 | 24.3 | **15.9** | 23.3 | 28.5 |
| 100 | 24.1 | **15.7** | 23 | 28.4 |

# 6 CONCLUSION

In this work, we introduced Federated Flow Matching (FFM), a novel framework for training flow-based generative models on decentralized data without compromising privacy. We identified the challenge of constructing effective couplings under federated constraints and proposed three algorithms with distinct advantages: FFM-vanilla provides a simple, stable baseline; FFM-LOT improves inference efficiency by leveraging local OT plans, but is sensitive to data heterogeneity; FFM-GOT enables faster sampling but can be sensitive to approximation errors. Our work points to several important directions for future research. These include reducing errors from dual potential approximations and improving communication efficiency. Besides, the federated flow matching framework we establish paves the way for integrating stronger privacy-preserving techniques, such as differential privacy or secure aggregation.

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

# A  ADDITIONAL THEORY AND PROOFS

## A.1  SEMI-DUAL OF THE SCHRÖDINGER BRIDGE PROBLEM

The Schrödinger bridge may be viewed as an entropic variant of optimal transport. Given a prior

$$R_\varepsilon(dx, dy) = q_0(dx)p_\varepsilon(dy \mid x),$$

the bridge is to seek the KL divergence between an estimation (posterior), i.e.,

$$\min_\pi \; D_{\mathrm{KL}}(\pi \| R_\varepsilon) \quad \text{subject to} \quad \Pi_x \# \pi = q_0, \; \Pi_y \# \pi = q_{1,\lambda}.$$

Duality introduces potentials $f$ on the source and $g$ on the target, yielding

$$\max_{f,g} \int f \, dq_0 + \int g \, dq_{1,\lambda} - \log \iint e^{f(x)+g(y)} R_\varepsilon(dx, dy).$$

Eliminating $f$ leads to a one-potential semi-dual:

$$\max_g \left\{ \int g \, dq_{1,\lambda} - \int \log \left( \int e^{g(y)} p_\varepsilon(dy \mid x) \right) q_0(dx) \right\}. \tag{11}$$

This has the same structure as the OT semi-dual, but with the hard inequality replaced by a log-sum-exp smoothing from the prior kernel. The first term separates across clients,

$$\int g \, dq_{1,\lambda} = \sum_{i=1}^n \lambda_i \int g \, dq_1^i,$$

so the formulation is naturally implementable in a federated setting, and as $\varepsilon \to 0$, the smoothing disappears and the dual potentials converge to those of the classical Kantorovich problem

## A.2  A USEFUL LEMMA

**Lemma 2 (Convexity under mixing).** *For cost $c(x, y) = \|x - y\|^p$ with $p \geq 1$ and measures $\mu, \nu_1, \ldots, \nu_n \in \mathcal{P}_p(\mathbb{R}^m)$ with weights $\lambda \in \Delta^{n-1}$,*

$$\mathcal{W}_p^p\Big(\mu, \sum_{i=1}^n \lambda_i \nu_i\Big) \leq \sum_{i=1}^n \lambda_i \, \mathcal{W}_p^p(\mu, \nu_i). \tag{12}$$

*Equality holds if and only if a convex combination of optimal couplings from $\mu$ to $\nu_i$ is itself optimal for $\big(\mu, \sum_{i=1}^n \lambda_i \nu_i\big)$.*

## A.3  PROOF OF LEMMA 2

Choose optimal $\gamma_i \in \Pi(\mu, \nu_i)$ and set $\bar{\gamma} = \sum_{i=1}^n \lambda_i \gamma_i$, $\bar{\nu} = \sum_{i=1}^n \lambda_i \nu_i$. Linearity of push-forwards gives $\mathrm{proj}_x \# \bar{\gamma} = \mu$ and $\mathrm{proj}_y \# \bar{\gamma} = \bar{\nu}$, hence $\bar{\gamma} \in \Pi(\mu, \bar{\nu})$. Therefore, we have

$$\mathcal{W}_p^p(\mu, \bar{\nu}) = \inf_{\gamma \in \Pi(\mu, \bar{\nu})} \iint_{\mathbb{R}^m \times \mathbb{R}^m} \|x - y\|^p \, d\gamma \leq \int \|x - y\|^p \, d\bar{\gamma}$$

$$= \sum_{i=1}^n \lambda_i \iint_{\mathbb{R}^m \times \mathbb{R}^m} \|x - y\|^p \, d\gamma_i$$

$$= \sum_{i=1}^n \lambda_i \, \mathcal{W}_p^p(\mu, \nu_i),$$

which proves (12). The inequality becomes an equality exactly when $\bar{\gamma}$ attains the infimum, i.e., when $\bar{\gamma}$ is optimal for $(\mu, \bar{\nu})$.

## A.4 PROOF OF THEOREM 1

By convexity of $\mathcal{W}_2^2$ under mixing (Lemma 2),

$$\mathcal{W}_2^2\Big(\pi^*, \sum_{i=1}^n \lambda_i \pi_i^*\Big) \leq \sum_{i=1}^n \lambda_i \, \mathcal{W}_2^2(\pi^*, \pi_i^*). \tag{13}$$

Fix $i$. Regard $\pi^*, \pi_i^*$ as measures on $\mathbb{R}^{2d}$ with the product quadratic cost

$$c_{\text{prod}}\big((x,y),(x',y')\big) = \|x-x'\|^2 + \|y-y'\|^2.$$

Couple them via the common source by pushing forward $q_0$ through the operator

$$H : x \mapsto \big((x,T^*(x)),\ (x,T_i^*(x))\big).$$

Define the measure $\Gamma := H_{\#}q_0$. Then $(\text{pr}_1)_{\#}\Gamma = (\text{Id}, T^*)_{\#}q_0 = \pi^*$ and $(\text{pr}_2)_{\#}\Gamma = (\text{Id}, T_i^*)_{\#}q_0 = \pi_i^*$, so $\Gamma \in \Pi(\pi^*, \pi_i^*)$. Hence, we have

$$\mathcal{W}_2^2(\pi^*, \pi_i^*) \leq \iint c_{\text{prod}}\, d\Gamma = \int \|T^*(x) - T_i^*(x)\|^2 \, dq_0(x). \tag{14}$$

Using $\|T^*(x)\| \leq D$ and $\|T_i^*(x)\| \leq D$, we have pointwise

$$\|T^*(x) - T_i^*(x)\| \leq \|T^*(x)\| + \|T_i^*(x)\| \leq 2D,$$

so $\|T^*(x) - T_i^*(x)\|^2 \leq (2D)\,\|T^*(x) - T_i^*(x)\|$. Integrating and applying Cauchy–Schwarz on the probability space $(\mathbb{R}^d, q_0)$ gives

$$\int \|T^*(x) - T_i^*(x)\|^2 \, dq_0(x) \leq 2D \int \|T^*(x) - T_i^*(x)\| \, dq_0(x) \leq 2D \, \|T^* - T_i^*\|_{L^2(q_0)}. \tag{15}$$

By the quantitative $L^2$ stability of Monge maps (Mérigot et al., 2020, Thm. 3.1),

$$\|T^* - T_i^*\|_{L^2(q_0)} \leq C_0 \, \mathcal{W}_1\big(q_1, q_1^i\big)^{2/15} \leq C_0 \, \mathcal{W}_2\big(q_1, q_1^i\big)^{2/15}. \tag{16}$$

Combining (14)–(16) yields

$$\mathcal{W}_2^2(\pi^*, \pi_i^*) \leq 2C_0 D \, \mathcal{W}_2\big(q_1, q_1^i\big)^{2/15}.$$

Insert this bound into (13) and absorb constants to obtain the desired result.

## B ADDITIONAL ALGORITHMS

---

**Algorithm 4** DUALUPDATE

---

1: **for** client $i = 1, \ldots, n$ **do**
2:     Sample $x_0 \sim q_0$, $x_1 \sim q_1^i$
3:     Evaluate $f_\phi^c(x_1) \approx c(\bar{x}_0, x_1) - f(\bar{x}_0)$, where $\bar{x}_0$ is approximate solution of $\inf_{x_0} c(x_0, x_1) - f(x_0)$
4:     Form local loss $L_\phi^i \leftarrow f_\phi(x_0) + f_\phi^c(x_1)$
5:     Send gradient $\nabla_\phi L_\phi^i$ to server
6: **end for**
7: Server update $\phi \leftarrow \phi - \sum_{i=1}^n \lambda_i \nabla_\phi L_\phi^i$ and broadcast to clients

---

**Algorithm 5** Resample $(x_0, x_1) \sim \hat{\pi}_\phi^i$

---

1: **Input**: Candidate source samples $x_{0,k=1,\ldots,K}$, target samples $x_{1,j=1,\ldots,B}$, dual potential $f_\phi$.
2: **for** $j = 1$ **to** $B$ **do**
3:    Evaluate $A_{j,k} = c(x_{0,k}, x_{1,j}) - f_\phi(x_{0,k})$ for $k = 1, \ldots, K$
4:    $k^\star(j) \leftarrow \arg\min_{k \in \{1,\ldots,K\}} A_{j,k}$
5:    $\tilde{x}_{0,j} \leftarrow x_{0,k^\star(j)}$
6: **end for**
7: **Output**: $\{(\tilde{x}_{0,j}, x_{1,j})\}_{j=1}^B$

---

## C  EXPERIMENTAL DETAILS

### C.1  2D EXAMPLE

We present more details on the 2D example, including implementation details and additional experimental results. Most of the experiments were run on clusters using NVIDIA A100s.

#### C.1.1  IMPLEMENTATION DETAILS

**Flow matching model:** The vector field is approximated using a time-varying multilayer perceptron (MLP) adopted from Tong et al. (2023). For each method, the model is trained for 40,000 epochs with a batch size of 256, employing the Adam optimizer.

**Dual potential function:** The dual potential function $f_\phi$ is parameterized using a 3-layer MLP with 128 hidden units and ReLU activation functions. The c-transform $f_\phi^c$ is approximated by sampling 256 candidate source points from the prior distribution and selecting the point that minimizes $c(x_0, x_1) - f_\phi(x_0)$ for each target point $x_1$.

**Federated data setup:** We split the target distribution such that two clients hold disjoint subsets of the target samples. For the moon target distribution, client 1 contains the upper moon while client 2 contains the lower moon, as shown in Fig. 2 (a). For the 8Gaussian target distribution, we assign the four left-lower Gaussians to client 1 and the four right-upper Gaussians to client 2, ensuring a clear non-IID partition.

**Training details:** For FFM with local OT, we compute the optimal transport plan between mini-batches of 256 source and 256 target samples using the exact solver. We then sample from the resulting coupling to generate training pairs. For FFM with global OT, the dual potential $f_\phi$ is parameterized by a 3-layer MLP with 128 hidden units and ReLU activation functions. The $c(x_0, x_1)$ function is selected as $\frac{1}{2} \|x_0 - x_1\|^2$. We optimize the dual potential using the Adam optimizer with a learning rate of 0.0001, updating it every 5 training steps of the main flow matching model. The $c$-transform function $f_\phi^c$, which is defined as the minimization over source points, is approximated using the minimum among 256 candidate source points sampled from the prior distribution.

**Evaluation metrics:** We evaluate performance using the 2-Wasserstein distance between generated samples and the true target distribution. The true distribution is approximated by the empirical distribution with 10000 samples.

### C.1.2  ADDITIONAL RESULTS

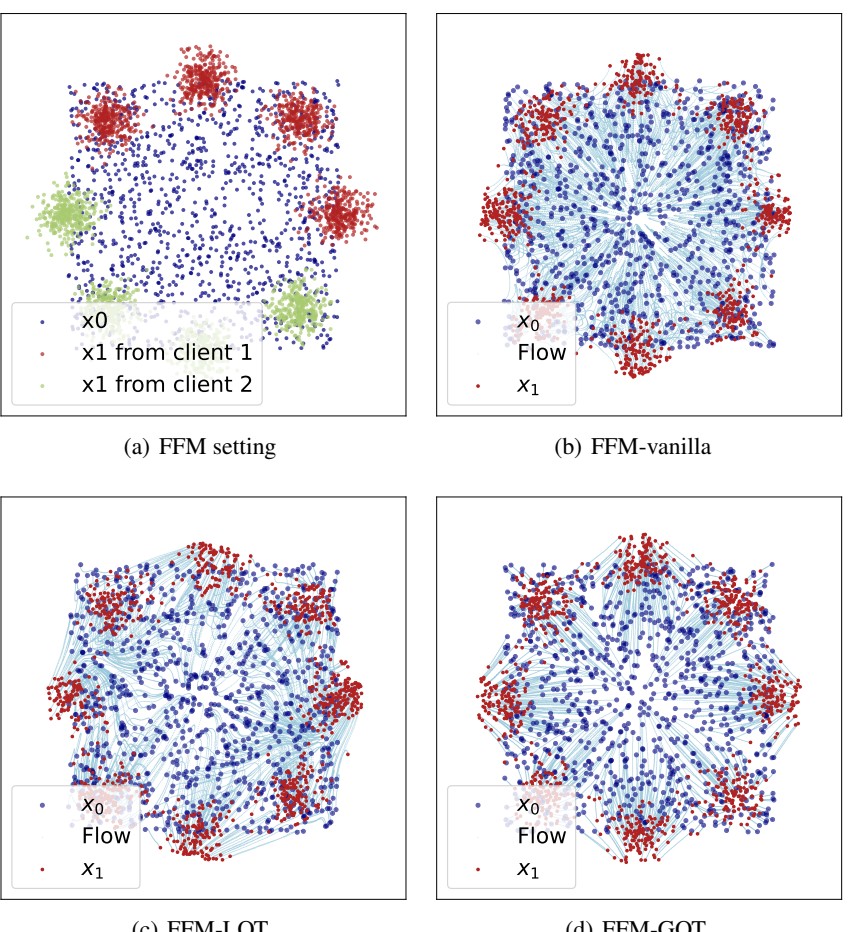

(a) FFM setting

(b) FFM-vanilla

(c) FFM-LOT

(d) FFM-GOT

Figure 4: Visualized trajectories learned by different FFM methods with uniform source distribution and 8Gaussian target distribution. FFM-vanilla yields curved trajectories. FFM-LOT improves straightness locally and FFM-GOT produces gloablly straight trajectories.

## C.2  CIFAR IMAGE GENERATION

We present more details on the CIFAR image generation, including implementation details and additional experimental results.

### C.2.1  IMPLEMENTATION DETAILS

**Flow matching models:**  The vector field model used in our experiments is trained using a U-Net architecture with a total of 35.75 million parameters adopted from Tong et al. (2023). The model uses a base channel dimension of 128, with channel multipliers $[1, 2, 2, 2]$ across four resolutions. It employs 2 residual blocks per resolution, attention mechanisms at the $16 \times 16$ resolution with 4 attention heads.

**Dual potential function:**  The global dual potential $f_\phi$ is parameterized by a convolutional neural network with a total of 1.15 million parameters. The network consists of six convolutional layers with spectral normalization and SiLU activations, progressively downsampling the input to a $4 \times 4$ resolution. A global average pooling layer is applied, followed by a linear layer to produce a scalar output.

**Federated data setup:** We simulate a federated learning environment with 3 clients. The CIFAR-10 training set is partitioned in a non-IID manner using a Dirichlet distribution with a concentration parameter $\alpha$. Client weights $\lambda_i$ are set proportional to their respective dataset sizes. In each training step, all 3 clients participate.

**Training details:** The U-Net parameters are optimized using Adam with a learning rate of $2 \times 10^{-4}$ and a linear warmup schedule for the first 5000 steps. The dual potential network is optimized using Adam with a learning rate of $10^{-4}$. Training runs for 400000 steps with a per-client batch size of 128. The training time of three methods is provided in Table 5. An exponential moving average of the U-Net parameters is maintained with a decay rate of 0.9999. The semi-dual objective is optimized using a quadratic cost function $c(x_0, x_1) = \frac{1}{2} \|x_0 - x_1\|^2$. The function value of $f_\phi^c$ is approximated through 5-step gradient descent with learning rate 0.5. The pairing of $(x_0, x_1)$ is performed through a memory-efficient resampling procedure: for each target sample $x_1$, the corresponding source point $x_0$ is selected from 128 candidate source samples by minimizing the cost $c(x_0, x_1) - f_\phi(x_0)$.

**Evaluation metric:** We evaluate sample quality using Fréchet Inception Distance (FID) calculated between 50000 generated samples and the CIFAR-10 test set. Measurements are taken at different numbers of function evaluations (NFE: 4, 10, 20, 50, 100) to assess inference efficiency. The inference time is reported in Table 6. We observe that the inference time is approximately proportional to the value of NFE.

Table 5: Training time for 400k steps on CIFAR using a single NVIDIA A100 GPU.

| Method | FFM-vanilla | FFM-LOT | FFM-GOT | Centralized OT-CFM |
|---|---|---|---|---|
| Training time ($\times 10^3$ s) | 138 | 141 | 189 | 55 |

Table 6: Inference time to generate 50000 CIFAR samples across varying NFE values on a single A100 GPU.

| NFE | 3 | 10 | 20 | 50 | 100 |
|---|---|---|---|---|---|
| Inference time (s) | 79 | 164 | 305 | 730 | 1437 |

### C.2.2 ADDITIONAL RESULTS

Table 7: FID of different methods on CIFAR dataset using dopri5 integration solver. FFM-LOT achieves superior performance, exceeding all other federated methods and even the centralized OT-CFM. This is attributed to its larger effective batch size: OT-CFM uses a batch size of 128, while FFM-LOT leverages a collective batch size of 384 (3 clients × 128).

| Method | FFM-vanilla | FFM-LOT | FFM-GOT | Centralized OT-CFM |
|---|---|---|---|---|
| FID | 4.37 | **3.57** | 8.35 | 3.85 |

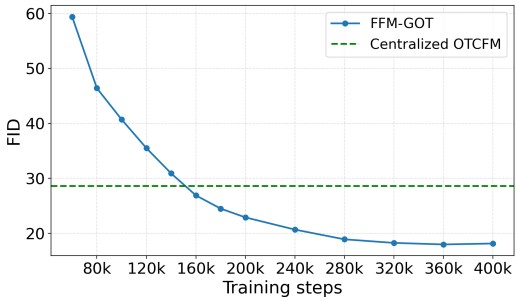

Figure 5: Comparison between our federated method FFM-GOT with the centralized method OT-CFM in Tong et al. (2023) at NFE=3. FFM-GOT surpasses the performance of OT-CFM after 160K training steps, and this performance gap widens significantly with further training.

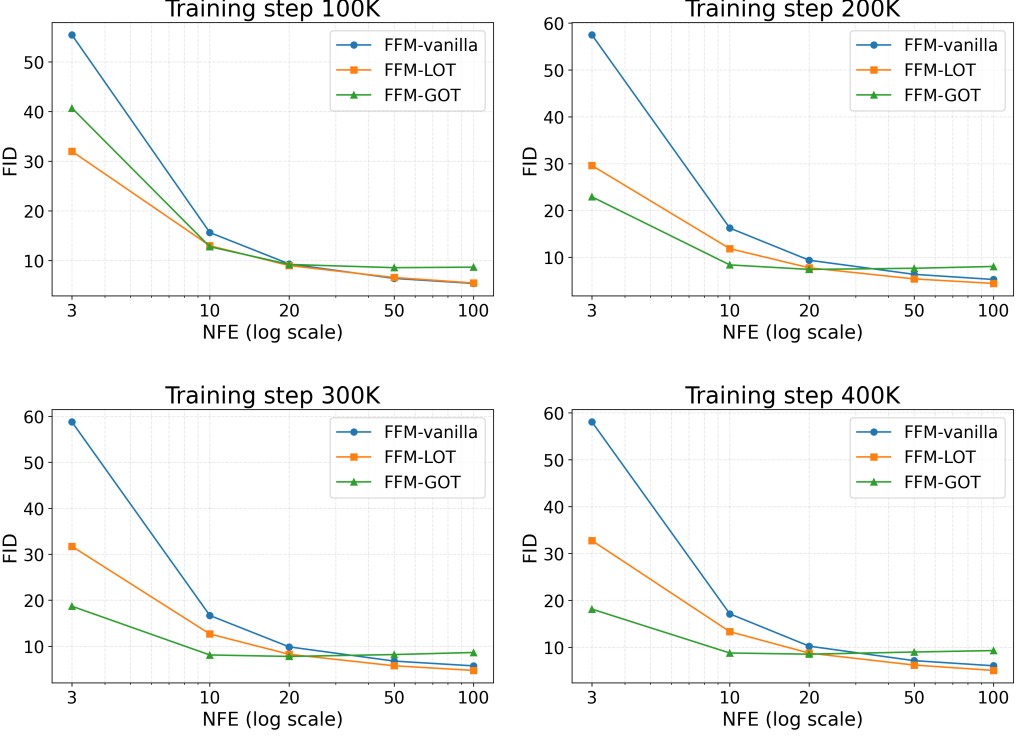

Figure 6: FID vs NFE of different methods on the CIFAR dataset at different training steps. FFM-GOT performs best when NFE is less than 20 after 200K training steps. When NFE is large, FFM-LOT achieves the best generation performance.

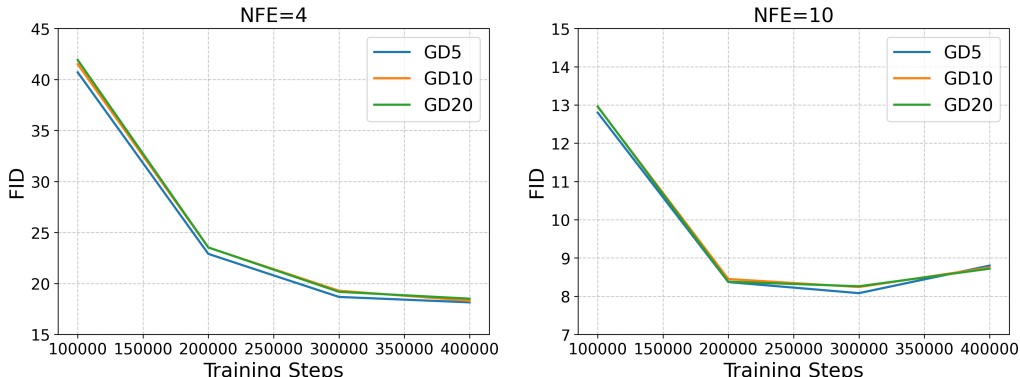

Figure 7: Generation performance of FFM-GOT with different gradient descent steps to solve $\inf_{x_0} c(x_0, x_1) - \phi(x_0)$. GDk denotes FFM-GOT with k gradient descent steps for k $= 5, 10, 20$. We observe that FFM-GOT is not sensitive to the number of gradient descent steps.

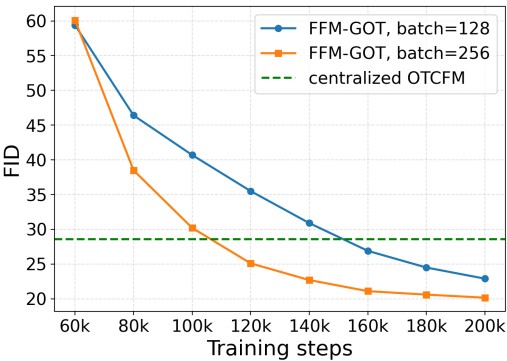

Figure 8: Generation performance of FFM-GOT with different batch sizes at NFE=3. Large batches yield better generation performance.

### C.3 ABLATION STUDY ON THE NUMBER OF GRADIENT-DESCENT STEPS AND CANDIDATE-POOL SIZE

We conduct ablation studies on both the number of gradient descent (GD) steps and the candidate pool size. Recall that in FFM-GOT we use a finite number of GD steps to approximate the solution of $\min_{x_0} c(x_0, x_1) - f_\phi(x_0)$. We train the models for 200K iterations with different numbers of GD steps: 2, 5, 10, and 20. As shown in Table 8, the number of GD steps has little effect on the resulting FID scores, and the performance is satisfactory with only 2 GD steps. We believe this is because the highly non-convex landscape of $f_\phi$ causes vanilla GD to get stuck in local minima, and additional steps offer limited benefit.

Table 8: FID results with different numbers of gradient descent steps

| gradient descent steps | NFE=3 | NFE=10 | NFE=20 | NFE=50 | NFE=100 |
|---|---|---|---|---|---|
| 2 | 23.54 | 8.43 | 7.33 | 7.51 | 7.83 |
| 5 | 22.90 | 8.37 | 7.41 | 7.65 | 8.04 |
| 10 | 23.52 | 8.45 | 7.39 | 7.71 | 8.11 |
| 20 | 23.53 | 8.38 | 7.22 | 7.61 | 7.98 |

We further investigate the impact of the candidate pool size. Recall that the candidate pool is a fixed set of $x_0$ used to draw $(x_0, x_1) \sim \hat{\pi}_\phi^i$ under the dual potential $f_\phi$. Due to computational constraints,

all models were trained for 200K rounds with pool sizes of 64, 128, 256, and 512. The training time was nearly identical across these settings (25 hours for 64 and 25.5 hours for 512). As shown in Table 9, larger pools allow for more accurate sampling from $\hat{\pi}_\phi^i$, improving flow straightness as evidenced by lower FID scores at NFE=3. However, it leads to slightly worse FID at higher NFEs. We attribute this to the gradient-descent-based approximation used to update $\hat{\pi}_\phi^i$, which introduces a bias towards the true global OT coupling. The resulting errors accumulate over more sampling steps, degrading ultimate performance. This reveals a trade-off: our method trades generation performance at high NFEs for flow straightness and lower complexity. To address the limitations of the gradient-descent strategy, future work will explore using a separate neural network to directly estimate the optimal solution (similar to GANs) to improve overall performance.

Table 9: FID results with different candidate pool sizes

| candidate pool size | NFE=3 | NFE=10 | NFE=20 | NFE=50 | NFE=100 |
|---|---|---|---|---|---|
| 64 | 26.1 | 7.9 | 6.2 | 6.3 | 6.7 |
| 128 | 24.9 | 8.7 | 7.5 | 7.7 | 8.0 |
| 256 | 20.5 | 9.2 | 8.7 | 9.2 | 9.6 |
| 512 | 18.6 | 10.1 | 10.3 | 11.3 | 11.7 |

### C.3.1 GENERATED CIFAR SAMPLES

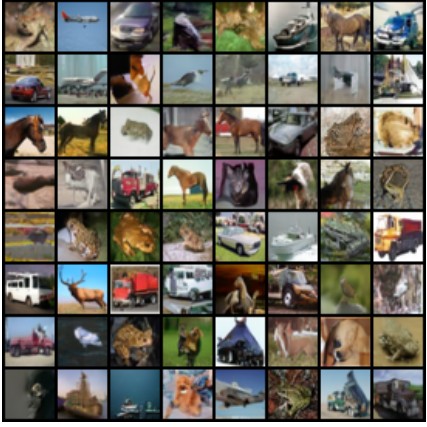

Figure 9: Generated samples from FFM-vanilla trained on CIFAR.

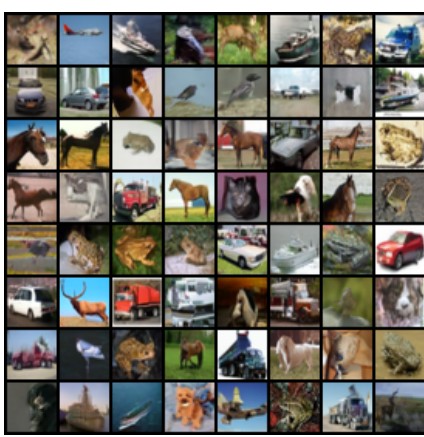

Figure 10: Generated samples from FFM-LOT trained on CIFAR.

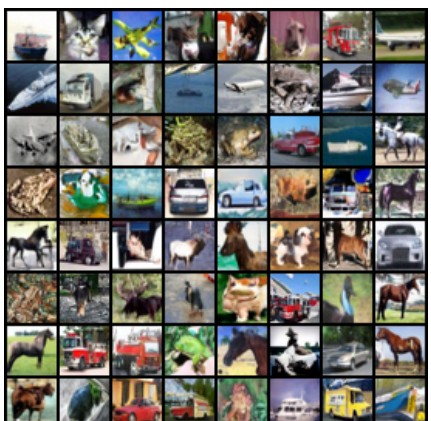

Figure 11: Generated samples from FFM-GOT trained on CIFAR.

## C.4 IMANGENET IMAGE GENERATION

### C.4.1 IMPLEMENTATION DETAILS

**Flow matching model:** We use a U-Net architecture with a base channel dimension of 128. The model consists of 4 resolution levels with channel multipliers [1,2,2,2], 2 residual blocks per resolution, and multi-head attention at the $16 \times 16$ resolution. The total number of parameters is 46.97 million.

**Dual potential function:** The global dual potential is parameterized by a convolutional neural network with 6 layers, spectral normalization, and SiLU activations. The total number of parameters is 1.15 million.

**Federated data setup:** The training dataset is derived from a 500-class subset of ImageNet, with 500 samples per class, resulting in a total of 250,000 images. Each image undergoes standardized preprocessing: it is resized to a resolution of $64 \times 64$ pixels, with random resized crops applied for data augmentation during training, followed by normalization to the range $[-1, 1]$. To emulate a realistic federated learning scenario with non-IID client distributions, the dataset is partitioned among 4 clients according to a Dirichlet distribution with concentration parameter $\alpha = 0.3$. Client weights $\lambda_i$ are assigned proportionally to their respective local dataset sizes.

**Training details:** The U-Net is optimized using Adam with a learning rate of $2 \times 10^{-4}$ while the dual potential network is optimized with Adam with a learning rate of $10^{-4}$. Training runs for 180,000 steps with a per-client batch size of 32. The training time is provided in Table 10. An exponential moving average of the U-Net parameters is maintained with a decay rate of 0.999. The semi-dual objective uses the quadratic cost $c(x_0, x_1) = \frac{1}{2} \|x_0 - x_1\|^2$. The function value of $f_\phi^c$ is approximated through 5-step gradient descent. For each $x_1$, the corresponding source sample $x_0$ is selected by finding the point among 32 candidate samples drawn from $q_0$ that minimizes the cost $c(x_0, x_1) - f_\phi(x_0)$. This procedure yields a pair $(x_0, x_1)$ that is approximately sampled from the global optimal transport plan $\hat{\pi}_\phi$.

**Evaluation metric:** We evaluate sample quality using FID calculated between 50000 generated samples and the Imagenet64-500 training dataset. Measurements are taken at different numbers of function evaluations to assess inference efficiency. The inference time is reported in Table 11.

Table 10: Training time for 200k steps on Imagenet64-500 using a single NVIDIA A100 GPU.

| Method | FFM-vanilla | FFM-LOT | FFM-GOT |
|---|---|---|---|
| Training time ($\times 10^3$ s) | 81 | 82 | 104 |

Table 11: Inference time to generate 50000 Imagenet64-500 samples across varying NFE values on a single A100 GPU.

| NFE | 4 | 10 | 20 | 30 |
|---|---|---|---|---|
| Inference time (s) | 82 | 159 | 290 | 676 |

### C.4.2 ADDITIONAL RESULTS ON IMAGENET64-10

We further evaluate the generation performance on Imagenet64-10, with only 10 classes of images from Imagenet. This is a more challenging task due to its limited size of only 9,346 training samples. The FID scores are presented in Table 12. The results show that FFM-LOT achieves the best performance across all NFE values, outperforming both FFM-vanilla and FFM-GOT. The overall higher FID scores across methods reflect the difficulty of learning a high-quality generative model from such a small and decentralized dataset. The particularly challenging conditions appear to magnify the approximation errors inherent in FFM-GOT's global OT approach, which requires sufficient

data to estimate the dual potential and optimal couplings accurately. In contrast, FFM-LOT proves more robust to limited data availability.

Table 12: FID scores across NFE values of Imagenet64-10 dataset using Euler integration.

| Method | NFE=3 | NFE=10 | NFE=20 | NFE=50 | NFE=100 |
|---|---|---|---|---|---|
| FFM-vanilla | 130.5 | 46.7 | 31.3 | 25.4 | 23.4 |
| FFM-LOT | **93.5** | **39.1** | **29.1** | **25.0** | **22.8** |
| FFM-GOT | 112.2 | 46.3 | 34.5 | 30.9 | 27.8 |

