# OpenReview forum: "Federated Flow Matching"
_ICLR.cc/2026/Conference — Submitted to ICLR 2026_

### Official Review · Reviewer_H4d7 · 2025-10-28

**Soundness:** 3
**Presentation:** 3
**Contribution:** 2
**Rating:** 4
**Confidence:** 3

**Summary:**

The paper introduces Federated Flow Matching (FFM), a framework for training flow-based generative models on decentralized data while preserving privacy. Unlike conventional flow matching, which assumes centralized datasets, FFM enables learning across multiple clients that keep their data local and only share model updates through federated averaging. The main challenge lies in defining effective couplings between a shared source distribution and an aggregated global target without directly accessing client data. To address this, the paper proposes three variants: FFM-Vanilla, which uses independent source–target pairings on each client and serves as a simple, privacy-preserving baseline but produces curved probability flows and slow inference; FFM-LOT, which computes local optimal transport (OT) plans per client to yield straighter trajectories and faster inference, though it suffers from sub-optimality under heterogeneous (non-IID) data; and FFM-GOT, which leverages the semi-dual formulation of OT to jointly learn a global dual potential that coordinates local couplings and approximates the true global OT plan, achieving globally straight flows and efficient inference at higher computational cost. The proposal has been experimentally evaluated.

**Strengths:**

1. The paper tackles a timely and important problem — extending flow matching to the federated learning setting — enabling privacy-preserving generative modeling without centralizing data.

2. It introduces a clear and systematic framework (FFM) with three variants (FFM-Vanilla, FFM-LOT, FFM-GOT), each addressing limitations of the previous one and providing a transparent progression in design.

3. The FFM-GOT method is novel, using the semi-dual formulation of optimal transport to coordinate global couplings across clients without sharing raw data, representing a significant conceptual and technical advance.

4. Experimental evaluation of the approach.

**Weaknesses:**

1. The paper presents FFM as a privacy-preserving framework, but to the best of my understanding the only privacy mechanism used is the standard federated averaging (FedAvg) exchange of model gradients. While this ensures that raw data remain local, it does not prevent potential information leakage through gradients. No formal threat model, differential privacy (DP) guarantees, or secure aggregation mechanisms are discussed. As a result, the privacy protection is qualitative rather than theoretically supported.

2. Theorem 1 rigorously quantifies how the mixture of local OT plans becomes sub-optimal under heterogeneous data, and experiments confirm that FFM-LOT’s performance degrades when the Dirichlet α decreases (α = 0.1). However, I was wondering if the paper should propose any mitigation strategies to handle non-IID settings, which are common in real federated scenarios.

3. The global OT variant depends on approximating the  c-transform via a small number of gradient-descent steps and a finite pool of candidate source samples. The paper acknowledges that these approximations accumulate and degrade performance at higher NFEs but does not provide a quantitative analysis of their impact or guidance on choosing the number of steps or candidate size.

4. The reported training times show significant overhead. However, the paper does not analyze communication rounds, data transfer size, or scalability with respect to the number of clients. These are critical aspects for federated systems, where communication efficiency is often a major bottleneck.

5. Experiments involve only three or four simulated clients on moderate-sized datasets (CIFAR-10 and ImageNet64-500). This setup does not reflect the scale or variability of practical federated learning, where tens or hundreds of clients may participate asynchronously. Evaluating FFM under such conditions would strengthen the evidence of robustness and scalability.

6. The experimental section compares FFM variants only against the centralized OT-CFM baseline. Although the related-work section discusses federated diffusion models and GAN-based methods, these are not included in the empirical comparison. Including at least one federated generative baseline would clarify the relative advantages and trade-offs of FFM.

7.  Although architectures and some hyperparameters are listed, certain implementation choices are missing. Public release of code would significantly improve reproducibility.

**Questions:**

1. Could you please clarify the specific privacy guarantees of FFM beyond keeping data local — for example, whether gradient leakage, differential privacy, or secure aggregation have been considered or analyzed?

2. Given that FFM-LOT degrades under high heterogeneity (e.g., α = 0.1), do you plan to incorporate any mitigation strategy to improve robustness in non-IID settings?

3. Can you provide quantitative results or ablations showing how the number of gradient-descent steps and candidate-pool size in the
c-transform approximation affect generation quality and convergence?

4. Could the you please report the number of communication rounds, data transmitted per round, and discuss how the computational and communication cost would scale as the number of clients increases?

5. Since experiments involve only three or four simulated clients, do you plan to evaluate FFM on larger-scale or more realistic federated settings with many heterogeneous clients or partial participation?

6. Why were federated generative baselines not included in the comparisons, and could you discuss how FFM is expected to perform relative to such methods?

7. Will the authors release code or provide additional implementation details (e.g., OT solver parameters, communication intervals, dual-potential updates) to ensure full reproducibility of the reported results?

---

> ### Author Response · Authors · 2025-11-22
>
> We sincerely thank the reviewer for insightful feedback allowing us to improve our work. We made additional experiments to address the concerns of the reviewer and outline responses below.
>
> > 1. No formal threat model, differential privacy (DP) guarantees, or secure aggregation mechanisms are discussed
>
> We acknowledge that the current version of our privacy mechanism is standard: it assumes trusted server, provides no formal differential privacy guarantees, and can be prone to leakage information. However, our primary contribution lies in establishing foundational methods for training flow-based models in a federated setting, which is a crucial step from centralized to decentralized learning. This foundation opens many directions for future work and enables the integration of stronger privacy-preserving techniques, such as differential privacy or secure aggregation, into our proposed algorithms. We believe all our methods are compatible with such advanced privacy mechanisms, and we leave their incorporation and analysis to future work.
>
> > 2. Mitigation strategy to handle non-IID settings for FFM-LOT
>
> Thanks for raising this point. In our view, the degradation under non-IID data stems from a fundamental limitation of approximating the global optimal transport coupling by mixing local OT couplings. This approximation is inherently inflexible and, in general, cannot be guaranteed to be effective when client distributions are heterogeneous. While problem-specific heuristics could partially mitigate the issue in certain settings, they do not provide a principled or broadly applicable solution. This is precisely why we propose FFM-GOT, which directly approximates the global OT coupling via a semi-dual formulation and is therefore better suited to handling non-IID federated scenarios.
>
> > 3. Ablations on the number of gradient-descent steps and candidate-pool size in the c-transform approximation
>
> Thanks for the valuable feedback. We have added ablation studies on both the number of gradient descent (GD) steps and the candidate pool size. Recall that in FFM-GOT we use a finite number of GD steps to approximate the solution of $\min_{x_0} c(x_0,x_1) - f_{\phi}(x_0)$. We additionally train the models for 200K iterations (due to limited compute) with different numbers of GD steps: 2, 5, 10, and 20. As shown in Table 1 below, the number of GD steps has little effect on the resulting FID scores, and the performance is satisfactory with only 2 GD steps. We believe this is because the highly non-convex landscape of $f_{\phi}$ causes vanilla GD to get stuck in local minima, and additional steps offer limited benefit.
>
> Table 1. FID results with different numbers of gradient descent steps
>
> | Gradient descent steps | NFE=3 | NFE=10 | NFE=20 | NFE=50 | NFE=100 |
> |------------------------|-------|--------|--------|--------|---------|
> | 2                      | 23.54 | 8.43   | 7.33   | 7.51   | 7.83    |
> | 5                      | 22.90 | 8.37   | 7.41   | 7.65   | 8.04    |
> | 10                     | 23.52 | 8.45   | 7.39   | 7.71   | 8.11    |
> | 20                     | 23.53 | 8.38   | 7.22   | 7.61   | 7.98    |
>
> We further investigate the impact of the candidate pool size. Recall that the candidate pool is a fixed set of $x_0$ used to draw $(x_0,x_1)\sim \hat{\pi}^i_{\phi}$ under the dual potential $f_{\phi}$. Due to computational constraints, models were trained for 200K rounds with pool sizes of 64, 128, 256, and 512. The training time was nearly identical across these settings (25 hours for 64 and 25.5 hours for 512). As shown in Table 2 below, larger pools allow for more accurate sampling from $\hat{\pi}^i_{\phi}$, improving flow straightness as evidenced by lower FID scores at NFE=3. However, it leads to slightly worse FID at higher NFEs. We attribute this to the gradient-descent-based approximation used to update $\hat{\pi}^i_{\phi}$, which introduces a bias towards the true global OT coupling. The resulting errors accumulate over more sampling steps, degrading ultimate performance. To address the limitations of the gradient-descent strategy, future work will explore using a separate neural network to directly estimate the optimal solution (similar to GANs) to improve overall performance. We will clarify these in the revised manuscript.
>
> Table 2. FID results with different candidate pool sizes
>
> | Candidate pool size | NFE=3 | NFE=10 | NFE=20 | NFE=50 | NFE=100 |
> |---------------------|-------|--------|--------|--------|---------|
> | 64                  | 26.1  | 7.9    | 6.2    | 6.3    | 6.7     |
> | 128                 | 24.9  | 8.7    | 7.5    | 7.7    | 8.0     |
> | 256                 | 20.5  | 9.2    | 8.7    | 9.2    | 9.6     |
> | 512                 | 18.6  | 10.1   | 10.3   | 11.3   | 11.7    |

---

> > ### Author Response · Authors · 2025-11-22
> >
> > > 4. Report the number of communication rounds, data transmitted per round, and discuss how the computational and communication cost would scale as the number of clients increases.
> >
> > Thanks for the valuable feedback. Here we clarify the communication and computation costs in one representative setting. We consider 32 clients in total; at each round, 3 clients are randomly selected to perform local training (batch size 128) and communicate with the server. Training runs for 200K rounds, giving a participation rate of 9.4\%. Let $|\theta|$ be the number of model parameters. In each round, every active client uploads its local model (or update) and downloads the global model, so with 3 active clients the data transmitted per round is about 6$|\theta|$ parameters. Thus, the communication cost scales linearly with the number of active clients per round. Note that our method is compatible with standard communication-efficient techniques (e.g., compression or quantization) to further reduce this overhead.
> >
> > The computational cost per round scales with the number of active clients and their local batch sizes n. For FFM-LOT, we solve one optimal transport (OT) problem per batch, with complexity cubic in n (or quadratic in n when using Sinkhorn). For FFM-GOT, the original formulation requires solving n non-convex optimization problems, but after relaxing this to a few gradient steps, the complexity becomes linear in n. Note that in all our algorithms, communication and computation per round are independent of the total number of clients. Empirically, the training time of FFM-GOT with 32 clients (3 active per round) is very close to that with only 3 clients (all active), and their results are similar (see the results with 32 clients in Reply 5). This indicates that, as long as the number of active clients per round is fixed, our method scales well to a large total number of clients.
> >
> > > 5. Limited number of clients
> >
> > Following your advice, we have evaluated our methods with $n=32$ clients to demonstrate scalability. In each iteration, 3 clients are randomly selected to compute gradients and communicate with the server. The per-iteration training time remains comparable to the setup with fewer clients, as it is primarily determined by the number of clients participating per round. Below, we report the FID and Inception scores for Dirichlet parameters $\alpha=0.5$ and $\alpha=0.1$, and the results are consistent with our previous findings. The strategy of selecting a subset of clients per round accelerates training without degrading performance, confirming that our methods scale effectively with the number of clients.
> >
> > Table 3. FID values on CIFAR with 32 clients
> >
> > | NFE | FFM-vanilla (α = 0.5) | FFM-LOT (α = 0.5) | FFM-GOT (α = 0.5) | FFM-vanilla (α = 0.1) | FFM-LOT (α = 0.1) | FFM-GOT (α = 0.1) | OT-CFM (Centralized) |
> > |-----|------------------------|-------------------|-------------------|------------------------|-------------------|-------------------|----------------------|
> > | 3   | 53.50                 | 31.48            | **23.40**         | 54.25                 | 30.92            | **25.42**         | 28.60               |
> > | 10  | 14.29                 | 12.33            | **8.78**          | 14.53                 | 12.09            | **9.32**          | 12.07               |
> > | 20  | 8.18                  | 8.14             | **7.78**          | 8.33                  | 8.03             | **7.97**          | 8.09                |
> > | 50  | **5.47**              | 5.59             | 7.96              | **5.49**              | 5.63             | 8.12              | 5.57                |
> > | 100 | **4.47**              | 4.53             | 8.30              | **4.64**              | 4.72             | 8.43              | 4.55                |
> >
> > Table 4. Inception scores on CIFAR with 32 clients
> >
> > | NFE | FFM-vanilla (α = 0.5) | FFM-LOT (α = 0.5) | FFM-GOT (α = 0.5) | FFM-vanilla (α = 0.1) | FFM-LOT (α = 0.1) | FFM-GOT (α = 0.1) | OT-CFM (Centralized) |
> > |-----|------------------------|-------------------|-------------------|------------------------|-------------------|-------------------|----------------------|
> > | 3   | 1.94                  | 2.44             | **3.08**          | 1.92                  | 2.45             | **3.02**          | 9.01                |
> > | 10  | 3.42                  | 3.79             | **4.71**          | 3.40                  | 3.81             | **4.63**          | 9.00                |
> > | 20  | 4.37                  | 4.54             | **5.40**          | 4.37                  | 4.57             | **5.32**          | 9.02                |
> > | 50  | 4.99                  | 5.09             | **5.90**          | 4.98                  | 5.12             | **5.80**          | 9.00                |
> > | 100 | 5.21                  | 5.22             | **5.99**          | 5.20                  | 5.29             | **5.91**          | 8.98                |

---

> > > ### Author Response · Authors · 2025-11-22
> > >
> > > > 6. Why were federated generative baselines not included in the comparisons and how FFM is expected to perform relative to such methods?
> > >
> > > We thank the reviewer for raising these points. We did not include federated generative baselines in the main comparisons because the comparison is not straightforward and, in our view, less informative, for two main reasons. First, the learning objectives are different. Existing federated diffusion models aim to “learn a flow” that produces high-quality samples, but they do not explicitly target flow straightness. In contrast, FFM is specifically designed to “learn the straight flow”, which is more challenging. Our experiments are therefore tailored to assess whether FFM improves flow straightness and, in turn, sample quality at low NFEs. The main motivations of prior federated generative works, such as communication efficiency [R2] or convergence guarantees [R1, R3], are largely orthogonal to this goal. Second, these methods use different model architectures and training setups (e.g., backbone networks, parameter counts, and training budgets), making a one-to-one empirical comparison difficult to interpret.
> > >
> > > That said, we can still provide a rough numerical comparison based on reported results. On CIFAR, the best FID reported in [R3] is 6.16 and in [R1] is 16.35, whereas FFM achieves FID < 5. At low NFEs, since existing federated diffusion models are not designed to promote flow straightness, we expect FFM to obtain substantially better FID.
> > >
> > > > 7. Release code?
> > >
> > > We have included the key implementation details in the paper and would be happy to provide any additional information if specified. We plan to release the code publicly after the paper is accepted.
> > >
> > > [R1] Zihao Peng, et al. Federated Learning for Diffusion Models. 2025.
> > >
> > > [R2] Matthijs de Goede, et al. Training diffusion models with federated learning. 2024.
> > >
> > > [R3] Jayneel Vora, et al. FedDM: enhancing communication efficiency and handling data heterogeneity in federated diffusion models. 2024.

---

### Official Review · Reviewer_tdSr · 2025-10-29

**Soundness:** 3
**Presentation:** 2
**Contribution:** 3
**Rating:** 4
**Confidence:** 4

**Summary:**

The paper introduces “Federated Flow Matching” (FFM), a framework aimed at training generative models on decentralized data while ensuring privacy by not centralizing raw data. It presents three methodologies: FFM-vanilla, which provides a stable baseline but results in curved flows; FFM-LOT, which leverages local optimal transport plans for improved inference efficiency; and FFM-GOT, which approximates a global optimal transport plan for straighter flows, yielding better performance despite increased computational demands. Experiments demonstrate that FFM significantly enhances sample quality and inference speed compared to centralized methods, addressing critical challenges in privacy-preserving training across distributed data sources.

**Strengths:**

1. The paper formalizes the idea of performing flow matching in a federated learning setting, which is both timely and conceptually valuable given the growing interest in privacy-preserving generative modeling.

2. The analysis connecting federated training and optimal transport (OT) couplings provides a meaningful mathematical perspective on why decentralized training induces curved or inconsistent flows.

3. The progression from FFM-Vanilla -> FFM-LOT -> FFM-GOT is logically structured, with each variant addressing limitations of the previous one.

**Weaknesses:**

The reviewer's main concerns stem from the experimental result.

1. The number of clients used in the experiments (n = 2/3/4) is too small to represent a realistic federated learning scenario, let alone capture the challenges of non-IID data distributions.

2. The Imagenet64-500 experiment lacks detailed reporting (compared with Table 1), making it unclear how results were obtained and why some details are missing.

3. The proposed FFM-LOT and FFM-GOT methods show only limited empirical improvement compared to vanilla federated learning. In some cases, their performance is even worse than the baseline (Table 1). Moreover, while the methods perform better at certain NFE ranges (either larger or smaller), they underperform in the opposite setting compared with FFM-vanilla. This inconsistent behavior raises concerns about whether the added algorithmic complexity offers practical value in real-world federated scenarios.

4. Although the authors claim that FFM-GOT is suitable for few-step generation, the experimental results show that when the number of function evaluations (NFE) is small (e.g., 3–10), the sample quality drops sharply, making the comparison less meaningful. To better demonstrate the advantages of GOT in low-step regimes, it is recommended to include comparisons with rectified-flow models or other flow-matching variants that are specifically designed for efficient few-step generation.

5. The paper mainly reports FID as the evaluation metric. Including additional measures such as Inception Score (IS) or other quality/diversity metrics would provide a more comprehensive assessment. Moreover, in the CIFAR experiments, the federated setting surprisingly outperforms the centralized baseline, which contradicts the common understanding in federated learning. Adding more metrics would help clarify this phenomenon and reveal whether the observed FID improvement comes at other potential costs (e.g., reduced diversity or overfitting).

**Questions:**

1. Could the author include additional evaluation metrics such as Inception Score (IS) or diversity-related measures? This would help clarify why the federated version sometimes outperforms the centralized baseline and whether the FID improvement involves any trade-offs.

2. The experiments use only a few clients, which may not sufficiently represent a realistic federated learning environment. Could you test your method under a larger number of clients or with more heterogeneous (non-IID) data to validate scalability?

3. Could the author complete Table 2 by including additional comparisons—such as the centralized baseline and results with larger NFE values—similar to those presented in Table 1?

4. The advantages of FFM-LOT and FFM-GOT are currently unclear. Although GOT is presented as an improvement over LOT, the results show GOT has almost no benefit for high-FID generation. Could the authors consider using a few-step generative model in the federated training setting to better demonstrate the potential advantages of GOT?

---

> ### Author Response · Authors · 2025-11-22
>
> We sincerely thank the reviewer for the detailed review and constructive feedback allowing us to improve our work. To address your concerns about the experimental results, we made additional experiments and clarification for each point. Please find the details below.
>
> > Test methods under a large number of clients to validate scalability
>
> Following your advice, we have evaluated our methods in a federated setting with $n=32$ clients to demonstrate scalability. In each iteration, 3 clients are randomly selected to compute gradients and communicate with the server. The per-iteration training time remains comparable to the setup with fewer clients, as it is primarily determined by the number of clients participating per round. Below, we report the FID and Inception scores for Dirichlet parameters $\alpha=0.5$ and $\alpha=0.1$, which align with our previous findings. The strategy of randomly selecting a subset of clients per round accelerates training without sacrificing performance, confirming that our methods scale effectively with the number of clients.
>
> Table 1. FID values on CIFAR with 32 clients
>
> | NFE | FFM-vanilla (α = 0.5) | FFM-LOT (α = 0.5) | FFM-GOT (α = 0.5) | FFM-vanilla (α = 0.1) | FFM-LOT (α = 0.1) | FFM-GOT (α = 0.1) | OT-CFM (Centralized) |
> |-----|------------------------|-------------------|-------------------|------------------------|-------------------|-------------------|----------------------|
> | 3   | 53.50                 | 31.48            | **23.40**         | 54.25                 | 30.92            | **25.42**         | 28.60               |
> | 10  | 14.29                 | 12.33            | **8.78**          | 14.53                 | 12.09            | **9.32**          | 12.07               |
> | 20  | 8.18                  | 8.14             | **7.78**          | 8.33                  | 8.03             | **7.97**          | 8.09                |
> | 50  | **5.47**              | 5.59             | 7.96              | **5.49**              | 5.63             | 8.12              | 5.57                |
> | 100 | **4.47**              | 4.53             | 8.30              | **4.64**              | 4.72             | 8.43              | 4.55                |
>
> Table 2. Inception scores on CIFAR with 32 clients
>
> | NFE | FFM-vanilla (α = 0.5) | FFM-LOT (α = 0.5) | FFM-GOT (α = 0.5) | FFM-vanilla (α = 0.1) | FFM-LOT (α = 0.1) | FFM-GOT (α = 0.1) | OT-CFM (Centralized) |
> |-----|------------------------|-------------------|-------------------|------------------------|-------------------|-------------------|----------------------|
> | 3   | 1.94                  | 2.44             | **3.08**          | 1.92                  | 2.45             | **3.02**          | 9.01                |
> | 10  | 3.42                  | 3.79             | **4.71**          | 3.40                  | 3.81             | **4.63**          | 9.00                |
> | 20  | 4.37                  | 4.54             | **5.40**          | 4.37                  | 4.57             | **5.32**          | 9.02                |
> | 50  | 4.99                  | 5.09             | **5.90**          | 4.98                  | 5.12             | **5.80**          | 9.00                |
> | 100 | 5.21                  | 5.22             | **5.99**          | 5.20                  | 5.29             | **5.91**          | 8.98                |
>
> > Incomplete Table 2
>
> The main reason Table 2 was initially incomplete is that we were limited by time and compute before the deadline. We have now completed it (Table 3 below), including the results for the centralized baseline and for larger NFE values. The results align with our previous findings: FFM-GOT performs well at low NFEs, and FFM-LOT performs better than FFM-vanilla.
>
> Table 3. FID values on Imagenet
>
> | NFE | FFM-vanilla (Fed, Dirichlet α=0.3) | FFM-LOT (Fed, Dirichlet α=0.3) | FFM-GOT (Fed, Dirichlet α=0.3) | OT-CFM (Centralized) |
> |-----|------------------------------------|--------------------------------|--------------------------------|----------------------|
> | 4   | 64.3                               | 43.7                           | **39.4**                       | 65.2                 |
> | 10  | 29.2                               | **18.6**                       | 28.4                           | 29.1                 |
> | 20  | 24.8                               | **16.3**                       | 23.2                           | 28.7                 |
> | 50  | 24.3                               | **15.9**                       | 23.3                           | 28.5                 |
> | 100 | 24.1                               | **15.7**                       | 23.0                           | 28.4                 |

---

> > ### Author Response · Authors · 2025-11-22
> >
> > >  Limited empirical improvement of FFM-LOT and FFM-GOT compared to FFM-vanilla
> >
> > At low NFEs, we believe the empirical improvement is significant. When $\alpha=0.3$ and NFE=3 (Table 1 in the paper), FFM-LOT and FFM-GOT achieve FIDs of 29 and 18, respectively, corresponding to a 49\% and a 68\% improvement over FFM-vanilla (FID 57). At high NFEs, the performance gap narrows since all methods converge to high performance (FID<10). For FFM-LOT, its performance is generally superior to FFM-vanilla. While it is sensitive to client distribution heterogeneity, its performance at high NFEs is only worse than FFM-vanilla under severely skewed data distributions (e.g., Dirichlet $\alpha=0.1$). For FFM-GOT, its task is learning the straight flow, which is fundamentally more challenging than learning a standard flow (the task of FFM-vanilla). Given limited compute, FFM-GOT trades flow quality for flow straightness: It enhances straightness, yielding superior performance at low NFEs, but unavoidable approximation errors accumulate with larger NFEs, leading to worse performance at high NFEs. Note that, in low-dimensional tasks with sufficient compute, FFM-GOT achieves the best performance (Fig 3). In the future work, we will explore whether similar performance can be achieved in high-dimensional tasks with more efficient algorithms and compute.
> >     In summary, our work delivers practical value by demonstrating superior performance of FFM-LOT and FFM-GOT at low NFEs, and by characterizing their distinct properties, which enables informed method selection for real-world federated scenarios.
> >
> > > Compare with rectified flow and other flow matching variants for few-step generation
> >
> > Thank you for raising this point. Note that rectified flow is a centralized algorithm, and cannot be directly extended to the federated setting due to its reliance on the global data distribution. The comparison between centralized rectified flow and our federated algorithms is less meaningful since the centralized rectified flow does not address federated challenges and employs a different strategy for improving flow straightness. We include the baseline OT-CFM because it shares a similar strategy for enhancing flow straightness. Comparing our federated algorithms with this centralized approach helps illustrate whether our method effectively addresses the challenges introduced by the federated setting. Moreover, few-step generation with flow-matching models in a federated setting is inherently challenging and, to the best of our knowledge, has not been previously explored. Therefore, we believe that the observed quality drop at low NFEs is reasonable.
> >
> > > Inception Score
> >
> > Thanks for the valuable suggestion. We compute Inception Score (IS) on CIFAR as shown in Table 4 below. We observe that FFM-GOT achieves the highest IS at all tested NFEs, indicating its ability to produce more diverse images. Regarding the observed outperformance relative to the centralized algorithm, our interpretation (see also line 445) is that the overall effective sample size used across federated clients per iteration is larger than the single batch used in the centralized algorithm. In other words, the performance gain from having a larger sample size outweighs the performance degradation introduced by federation. We would make it clear in the revised draft.
> >
> > Table 4. Inception score on CIFAR with 3 clients
> >
> > | NFE | FFM-vanilla (α=0.3) | FFM-LOT (α=0.3) | FFM-GOT (α=0.3) | FFM-vanilla (α=0.1) | FFM-LOT (α=0.1) | FFM-GOT (α=0.1) | OT-CFM (Centralized) |
> > |-----|------------------------------------|---------------------------------|---------------------------------|-------------------------------------|--------------------------------|--------------------------------|----------------------|
> > | 3   | 1.90                               | 2.49                            | **3.08**                        | 1.96                                | 2.47                           | **3.02**                       | 9.01                 |
> > | 10  | 3.27                               | 3.84                            | **4.75**                        | 3.47                                | 3.82                           | **4.69**                       | 9.00                 |
> > | 20  | 4.15                               | 4.61                            | **5.47**                        | 4.42                                | 4.59                           | **5.41**                       | 9.02                 |
> > | 50  | 4.75                               | 5.11                            | **5.92**                        | 5.11                                | 5.11                           | **5.88**                       | 9.00                 |
> > | 100 | 4.96                               | 5.29                            | **6.10**                        | 5.29                                | 5.28                           | **5.98**                       | 8.98                 |

---

> > > ### Author Response · Authors · 2025-11-22
> > >
> > > > Unclear advantages of FFM-LOT and FFM-GOT
> > >
> > > FFM-LOT successfully enhances flow straightness, indicated by better generation performance at low NFEs. Its performance remains stable at high NFEs and is generally superior to FFM-vanilla. Although it is sensitive to client distribution heterogeneity, it underperforms FFM-vanilla at high NFEs only under severely skewed data partitions (dirichlet alpha=0.1). FFM-GOT provides a more significant enhancement in flow straightness, leading to a substantial performance improvement at low NFEs. For example, on low-dimensional tasks it achieves the best performance across all NFE values (Fig. 3). In the image generation task, although approximation errors are unavoidable due to limited compute, it achieves the best results on CIFAR when the NFE is below 20. Moreover, it consistently attains the highest Inception Score across all settings, indicating its ability to generate highly diverse images. We will incorporate these clarifications into the revised draft.

---

### Official Review · Reviewer_czNW · 2025-10-30

**Soundness:** 3
**Presentation:** 3
**Contribution:** 3
**Rating:** 4
**Confidence:** 3

**Summary:**

Based on my understanding, this work introduces Federated Flow Matching (FFM), an approach that extends classical federated optimization methods such as FedAvg and FedProx to flow-based generative models. Instead of averaging model parameters directly, FFM aligns local and global flow fields through a flow-consistency regularization term, aiming to stabilize training under non-IID data distributions. The proposed framework is evaluated on standard image generation benchmarks, showing improved convergence and generation quality compared with basic federated baselines.

**Strengths:**

The paper provides a coherent extension of federated optimization to generative models.
Extending federated learning to flow-based generation addresses an important gap, as most prior FL works have focused on discriminative tasks.
Specifically, the loss terms proposed in Section 3.1 and 3.2 are clearly formulated and effectively reduces client drift in non-IID settings.
Experimental results demonstrate stable convergence and better global consistency than FedAvg and FedProx, confirming that the method is functional and correctly implemented.

**Weaknesses:**

1. The conceptual novelty is limited, as the framework closely mirrors FedAvg and FedProx.
The proposed flow-consistency term serves a similar role to the proximal regularization used in FedProx—both aim to prevent local updates from drifting too far from the global model under non-IID data.
The main difference lies in applying this constraint to function space (flow fields) rather than parameter space, which is a natural and technically reasonable extension but not a new optimization paradigm.
Furthermore, the paper lacks theoretical analysis (e.g., convergence or stability proofs) and does not clearly articulate whether operating in flow space introduces fundamentally new behaviors compared with existing regularized FedAvg variants.
Overall, the contribution feels incremental.
This is my understanding; if I have misunderstood the intended contribution, I welcome clarification in the rebuttal.

2. Experiments are limited to small-scale 2D image generation tasks.
The claim of scalability would be more convincing with results on larger or more diverse generative tasks, such as conditional or video generation.
Moreover, the empirical improvements over strong baselines remain modest, showing better stability but not clear superiority.

3. The number of clients is set to only 3 for CIFAR-10 and 4 for ImageNet64-500 (Section 4.1). This setting seems too small to represent realistic federated scenarios and makes the scalability claim less convincing. It would be helpful if the authors could justify this choice or report results under larger client counts.
“We simulate a federated learning environment with 3 clients. The CIFAR-10 training set is partitioned in a non-IID manner using a Dirichlet distribution with a concentration parameter α. Client weights λᵢ are set proportional to their respective dataset sizes. In each training step, all 3 clients participate.”
“We further evaluate generative performance on ImageNet64-500, a more challenging benchmark constructed from 500 ImageNet classes, each containing 500 samples, yielding a total of 250,000 images. We simulate n = 4 clients.”

The equation on page 5 is missing a reference number, by the way.

**Questions:**

Please check Weaknesses for details.

---

> ### Author Response · Authors · 2025-11-22
>
> We sincerely thank the reviewer for constructive comments allowing us to improve our work. We added additional experiments to address your concerns and outline responses below.
>
> > Limited conceptual novelty and incremental contribution.
>
> We would like to clarify and restate our contribution here. We are the first to explore training flow matching models in a federated setting. The key challenge, which distinguishes it from standard federated learning (FL), is a fundamental conflict: learning a straight flow is an inherently global objective relying on the entire data distribution, yet FL only allow local data views. This challenge crystallizes into a specific mathematical problem: how to design and implement the global coupling $\pi \in \Pi(q_0,q_{1,\lambda})$ for the training objective (4) such that it is realizable in a federated setting while still promoting straight flows. **Our contributions lie in novel designs and implementations of the coupling $\pi$ to achieve this goal.** Specifically, we propose FFM-LOT and FFM-GOT, which use local and global Optimal Transport (OT), respectively. Both methods improve flow straightness, as shown in the experiments. Notably, FFM-GOT is based on a key novel insight: the semi-dual formulation of OT is inherently compatible with federated optimization. We implement this by training a neural network to represent the semi-dual solution via FedAvg. In summary, we present a novel framework to formulate and solve the fundamental coupling problem for federated flow matching. Since this coupling problem is not existing in traditional FL, we believe our contributions are not incremental.
>
> Regarding the optimization paradigm, it is not the primary contribution of this work. All our methods use FedAvg as an update strategy, and are flexible to use other strategies such as FedProx or the version combined with differential privacy.
>
> Regarding the convergence analysis, the cases for FFM-vanilla and FFM-LOT are relatively straightforward under standard smoothness assumptions, as their training losses decouple across clients. In contrast, FFM-GOT presents a significant analytical challenge because its training objective is coupled across clients through global OT, which is inherently complex. While a theoretical analysis for FFM-GOT remains an open and non-trivial problem, we have empirically demonstrated that all three algorithms converge stably in practice.
>
> Regarding the flow-consistency term, we are not aware of its meaning because it was not used in our paper. We would appreciate it if you could provide a brief clarification so we can discuss it further.
>
> > Limited to small-scale 2D image generation tasks
>
> Thanks for mentioning the diverse tasks. We evaluated our methods on the standard public benchmarks of CIFAR and ImageNet. The computational demand for this evaluation was substantial: a single run for one parameter setting requires approximately 50 hours on an A100 GPU for 400K training steps. In total, our reported results represent over 2000 GPU hours. Given limited compute, we leave the extension to more complex domains, such as video generation, to future work.
>
> > Modest empirical improvements over baselines
>
> Since there is no established method for training flow-matching models in a federated setting, we assume that by “baselines” you are referring to the centralized OT-CFM and FFM-vanilla methods. We believe the empirical improvement at low NFEs is significant. As an example using CIFAR, when $\alpha=0.3$ and NFE=3, FFM-GOT achieves an FID of 18.15, which is a 68\% improvement over FFM-vanilla (FID 57), and notably a 36\% improvement over even the centralized algorithm (FID 28). At high NFEs, the performance gap is expected to narrow since all methods converge to high performance (FID<10 in Table 1).

---

> > ### Author Response · Authors · 2025-11-22
> >
> > > Limited number of clients
> >
> > Thanks for the insightful feedback. Following your suggestion, we have evaluated our methods in a federated setting with $n=32$ clients to demonstrate scalability. In each iteration, 3 clients are randomly selected to compute gradients and communicate with the server. The per-iteration training time remains comparable to the setup with fewer clients, as it is primarily determined by the number of clients participating per round. Below, we report the FID and Inception scores for Dirichlet parameters $\alpha=0.5$ and $\alpha=0.1$. The results align with our previous findings, confirming that our methods scale effectively with the number of clients.
> >
> > Table 1. FID values on CIFAR with 32 clients
> >
> > | NFE | FFM-vanilla (α = 0.5) | FFM-LOT (α = 0.5) | FFM-GOT (α = 0.5) | FFM-vanilla (α = 0.1) | FFM-LOT (α = 0.1) | FFM-GOT (α = 0.1) | OT-CFM (Centralized) |
> > |-----|------------------------|-------------------|-------------------|------------------------|-------------------|-------------------|----------------------|
> > | 3   | 53.50                 | 31.48            | **23.40**         | 54.25                 | 30.92            | **25.42**         | 28.60               |
> > | 10  | 14.29                 | 12.33            | **8.78**          | 14.53                 | 12.09            | **9.32**          | 12.07               |
> > | 20  | 8.18                  | 8.14             | **7.78**          | 8.33                  | 8.03             | **7.97**          | 8.09                |
> > | 50  | **5.47**              | 5.59             | 7.96              | **5.49**              | 5.63             | 8.12              | 5.57                |
> > | 100 | **4.47**              | 4.53             | 8.30              | **4.64**              | 4.72             | 8.43              | 4.55                |
> >
> > Table 2. Inception scores on CIFAR with 32 clients
> >
> > | NFE | FFM-vanilla (α = 0.5) | FFM-LOT (α = 0.5) | FFM-GOT (α = 0.5) | FFM-vanilla (α = 0.1) | FFM-LOT (α = 0.1) | FFM-GOT (α = 0.1) | OT-CFM (Centralized) |
> > |-----|------------------------|-------------------|-------------------|------------------------|-------------------|-------------------|----------------------|
> > | 3   | 1.94                  | 2.44             | **3.08**          | 1.92                  | 2.45             | **3.02**          | 9.01                |
> > | 10  | 3.42                  | 3.79             | **4.71**          | 3.40                  | 3.81             | **4.63**          | 9.00                |
> > | 20  | 4.37                  | 4.54             | **5.40**          | 4.37                  | 4.57             | **5.32**          | 9.02                |
> > | 50  | 4.99                  | 5.09             | **5.90**          | 4.98                  | 5.12             | **5.80**          | 9.00                |
> > | 100 | 5.21                  | 5.22             | **5.99**          | 5.20                  | 5.29             | **5.91**          | 8.98                |

---

### Author Response · Authors · 2025-12-02

Dear Area Chair and reviewers,

We sincerely thank you for your time, thoughtful feedback, and engagement with our work. We have incorporated the feedback to improve our work thoroughly, with changes marked in blue in the updated submission. Here, we would like to summarize the discussion and make a few final comments to assist AC in evaluating our paper.

We begin by briefly summarizing our key contributions. This work presents the **first** exploration of training flow matching models in a federated setting. The key challenge, which distinguishes it from standard federated learning, is learning a **straight** flow from decentralized data.
This challenge is crystallized into a core mathematical problem concerning the coupling in global optimal transport.
Our contributions lie in novel algorithmic designs and implementations of this coupling that promote flow straightness while remaining practical in a federated setting. In particular, one of our proposed algorithms, FFM-GOT, is built on a key novel insight: the semi-dual formulation of optimal transport is inherently compatible with federated optimization. Experiments on image generation tasks illustrate the effectiveness and scalability of our algorithms.

Reviewers have a common concern about applicability in settings with more clients. In response, we conducted additional experiments with 32 clients, and the performance results align with our previous findings. Crucially, the training time stays efficient, as it mainly depends on the number of active clients per iteration, thereby demonstrating the scalability of our algorithms to large-scale federated settings. We have included these results in the updated paper.


**Reviewer czNW** (rating 4, confidence 3) appreciates the important problem setup, well-formulated technical approach, and experimental validation. Regarding their concern about our contribution, we believe it was initially misunderstood and we have provided clarification in both response and the updated paper. Regarding the scale of experiments, we believe CIFAR and ImageNet datasets represent standard benchmarks of appropriate scale to validate flow matching models, and the total GPU usage is ~2000 hours.

**Reviewer tdSr** (rating 4, confidence 4) appreciates the valuable problem formulation, insightful theoretical analysis, and well-structured methodology development. In response to their concerns about the experiments, we have conducted additional experiments with more clients, completed Table 2 for Imagenet, and reported additional evaluation metric (Inception Score). These additional experimental results further demonstrate the scalability and effectiveness of our methods. We have included all these results in the main body or appendix of the updated paper. Regarding their comment on benchmarking against rectified flow models, we believe a fair comparison is difficult because extending rectified flow to federated setting is not feasible. We also clarified the points about limited empirical improvement and the advantages of FFM-LOT and FFM-GOT in both response and updated paper.


**Reviewer H4d7** (rating 4, confidence 3) appreciates the timely and important problem, the systematic FFM framework, the novel conceptual and technical contribution of the FFM-GOT, and the experimental evaluation. They request more experimental results, including ablation studies on the c-transform approximation, communication cost analysis, and settings with more clients. We have provided all of these in both response and updated paper, and the new results further illustrate the scalability and effectiveness of our methods.
Regarding the comparison with federated diffusion models, our FFM methods achieve much better FID than their reported results.
Regarding the privacy mechanism, we clarified that we use vanilla FedAvg and emphasized that our framework **opens many directions** for incorporating stronger privacy-preserving techniques such as differential privacy and secure aggregation.
We also responded to the remaining questions about mitigation strategies and the release of code.



In conclusion, echoing **Reviewer H4d7**’s summary, we _introduce a systematic framework (FFM) with three variants, each addressing limitations of the previous one and offering a transparent progression; in particular, FFM-GOT is built on a novel insight that the semi-dual formulation of optimal transport is compatible with federated optimization_. We believe this work provides a novel connection between federated learning and generative flow matching, opening many future directions. **Although we did not receive further comments from the reviewers after our initial responses, we believe that our detailed responses and new results have addressed all of their concerns, and further strengthens the case for acceptance.**

Sincerely,

Authors

---

### Meta-Review · Area_Chair_3hqk · 2026-01-05

**Summary:**

The reviewers collectively expressed concerns primarily regarding the novelty of the proposed approach, its theoretical foundations, and the adequacy of the empirical evaluations.

(a) There is consensus that the paper’s contributions are incremental rather than fundamentally novel, as the proposed framework closely resembles existing federated optimization methods (e.g., FedAvg/FedProx), with the main distinction being a regularization applied in function/flow space instead of parameter space. Reviewers consistently question whether this change introduces qualitatively new behavior, noting the absence of heoretical analysis e.g., convergence, stability, or privacy guarantees) to justify the claimed advantages.

(b) A second major concern is the limited experimental evaluation: all reviewers highlight the use of very small numbers of simulated clients (3 or 4), modest-scale image generation benchmarks, and relatively small empirical gains that are sometimes inconsistent or inferior to baselines. The scalability and practicality of the method in realistic federated settings — with many heterogeneous clients, communication constraints, and non-IID data — remain insufficiently demonstrated.

(c) Reviewers also note gaps in evaluation rigor, including reliance on a single metric (FID), lack of federated generative baselines, and limited analysis of computational and communication overhead.

(d) Finally, claims around privacy preservation are viewed as qualitative and under-specified, with no formal threat model or guarantees provided.

**Reviewer Concerns:**

The authors provided additional experimental results, including Inception Score as an evaluation metric and experiments with a larger total number of clients (32 clients with 3 active per round). These additions partially address reviewer concerns regarding evaluation rigor and scalability in more realistic federated settings. The rebuttal also includes a more detailed analysis of computational and communication overhead, which helps ground the proposed method in practical deployment considerations.

However, several core issues remain insufficiently addressed. (a) Technical novelty and theoretical grounding: while the authors clarify that the main contribution is enforcing straight-flow learning in a federated learning setting, the necessity and theoretical implications of learning straight flows in this context are not formally justified. In particular, there is still no analysis of convergence, stability, or privacy guarantees, leaving these major concerns only partially resolved. (b) Scalability: although experiments with a larger total number of clients are included, the number of active clients per round remains fixed at three, which limits the strength of the scalability claims. Evaluating performance under varying numbers of active clients — even at small scales — would substantially strengthen the empirical evidence. (c) Computational overhead: the rebuttal provides conflicting signals regarding efficiency. On the one hand, the authors report that a single experimental run requires approximately 50 GPU hours on an A100, totaling over 2000 GPU hours (response to reviewer czNW); on the other hand, they state that the method scales well as long as the number of active clients is fixed (response to reviewer H4d7). This apparent inconsistency would benefit from a clearer and more systematic breakdown of computational and communication costs as a function of both total and active client counts.

**Reviewer Scores:**

I believe that all reviewers would likely maintain their original scores, as their primary concerns — particularly regarding theoretical guarantees, scalability, and the sufficiency of the empirical evaluation — were only partially addressed in the authors’ rebuttal.

---

### Decision · Program_Chairs · 2026-01-26

Reject